# Piwi reduction in the aged niche eliminates germline stem cells via Toll-GSK3 signaling

Kun-Yang Lin [1,2,3], Wen-Der Wang [4], Chi-Hung Lin[3], Elham Rastegari[3], Yu-Han Su[3], Yu-Tzu Chang [3], Yung-Feng Liao[3], Yi-Chieh Chang [5], Haiwei Pi[5], Bo-Yi Yu[6], Shu-Hwa Chen[6], Chung-Yen Lin [6], Mei-Yeh Lu [7], Tsu-Yi Su [8], Fei-Yang Tzou[8], Chih-Chiang Chan [8] & Hwei-Jan Hsu [1,3,9 ✉]

Transposons are known to participate in tissue aging, but their effects on aged stem cells remain unclear. Here, we report that in the *Drosophila* ovarian germline stem cell (GSC) niche, aging-related reductions in expression of Piwi (a transposon silencer) derepress retrotransposons and cause GSC loss. Suppression of Piwi expression in the young niche mimics the aged niche, causing retrotransposon depression and coincident activation of Toll-mediated signaling, which promotes Glycogen synthase kinase 3 activity to degrade β-catenin. Disruption of β-catenin-E-cadherin-mediated GSC anchorage then results in GSC loss. Knocking down *gypsy* (a highly active retrotransposon) or *toll*, or inhibiting reverse transcription in the *piwi*-deficient niche, suppresses GSK3 activity and β-catenin degradation, restoring GSC-niche attachment. This retrotransposon-mediated impairment of aged stem cell maintenance may have relevance in many tissues, and could represent a viable therapeutic target for aging-related tissue degeneration.

[1] Molecular and Biological Agricultural Sciences Program, Taiwan International Graduate Program, National Chung Hsing University and Academia Sinica, Taipei 11529, Taiwan. [2] Graduate Institute of Biotechnology, National Chung Hsing University, Taichung 40227, Taiwan. [3] Institute of Cellular and Organismic Biology, Academia Sinica, Taipei 11529, Taiwan. [4] Department of BioAgricultural Sciences, National Chiayi University, Chiayi City 60004, Taiwan. [5] Graduate Institute of Biomedical Sciences, College of Medicine, Chang Gung University, Kweishan, Taoyuan, Taiwan. [6] Institute of Information Science, Academia Sinica, Taipei 11529, Taiwan. [7] Biodiversity Research Center, Academia Sinica, Taipei 11529, Taiwan. [8] Graduate Institute of Physiology, College of Medicine, National Taiwan University, Taipei 10617, Taiwan. [9] Biotechnology Center, National Chung Hsing University, Taichung 40227, Taiwan. ✉email: cohsu@gate.sinica.edu.tw

Transposable genetic elements (transposons) are abundant in eukaryotic genomes[1–3]. In particular, retrotransposons, which replicate by reverse transcription, occupy 16% of fly and 40% of human genomes[2,4,5]. Despite their genomic abundance, transposons are effectively silenced by Piwi-interacting RNAs (piRNAs) at transcriptional and post-transcriptional levels in the germline to protect intact genetic inheritance[6,7]. However, transposon silencing is known to be attenuated in aged tissues[8–10], where stem cells are frequently lost[11]. It is not clear, however, whether transposon derepression contributes to age-dependent stem cell decline.

## Results

**Aging reduces Piwi expression in the GSC niche.** Our group and others have used the *Drosophila* ovary to study the decline of GSC number with age[12,13]. In this model system, GSCs are located at the anterior tip of the germarium and form direct contacts with cap cells (CpCs, the major GSC niche component), which are adjacent to the terminal filament (TF) and anterior escort cells (ECs) (Fig. 1a); together, these cells form the GSC niche[14]. A single GSC gives rise to a cystoblast, which later generates a functional oocyte[15]. Interestingly, we found that Piwi was highly expressed in every cell of the young germarium, except the TF[16], but its expression was nearly absent in CpCs of aged flies (Fig. 1b and Supplementary Fig. 1a–g). To examine the role

of Piwi in adult CpCs, we individually used two independent *UAS-RNAi* lines (*piwi^{RNAi-1}* and *piwi^{RNAi-2}*) with the niche-specific driver, *bab1-GAL4*[17], under the control of temperature sensitive *GAL80^{ts}*[18]. Flies expressing *piwi^{RNAi}* were cultured at 18 °C to suppress GAL4 activity during developmental stages, and were switched to 29 °C after eclosion to degrade GAL80^{ts} and activate GAL4. Piwi protein expression remained in CpCs of 1-week-old *bab1 > piwi^{RNAi-1}* and *bab1 > piwi^{RNAi-2}* germaria (data not shown), but was nearly absent in *piwi*-knockdown (KD) CpCs at 2 weeks as compared to controls (*bab1 > gfp^{RNAi}*) (Fig. 1c). We then examined the number of GSCs in newly eclosed (Day 1), 2- and 5-week-old *bab1 > piwi^{RNAi}* flies by observing their anteriorly anchored fusome (a membranous cytoskeletal structure) adjacent to CpCs[19]. As previously reported[12,13], the number of GSCs in control flies decreased with age (Fig. 1d and Supplementary Table 1). This decrease was accelerated in *piwi*-KD lines (Fig. 1d and Supplementary Table 1). Compared to newly eclosed germaria, 83% of GSCs remained in 2-week-old control germaria ($n = 108$); however, only 66 and 68% of GSCs were found in 2-week-old *bab1 > piwi^{RNAi-1}* ($n = 94$; $P < 0.001$) and *bab1 > piwi^{RNAi-2}* germaria ($n = 101$; $P < 0.001$), respectively. Although newly eclosed *bab1 > piwi^{RNAi-2}* flies carried higher baseline numbers of GSCs compared to *bab1 > piwi^{RNAi-1}* flies and controls (probably due to their different genetic backgrounds), the rate of GSC loss in *bab1 > piwi^{RNAi-2}* was comparable to *bab1 > piwi^{RNAi-1}* flies and higher

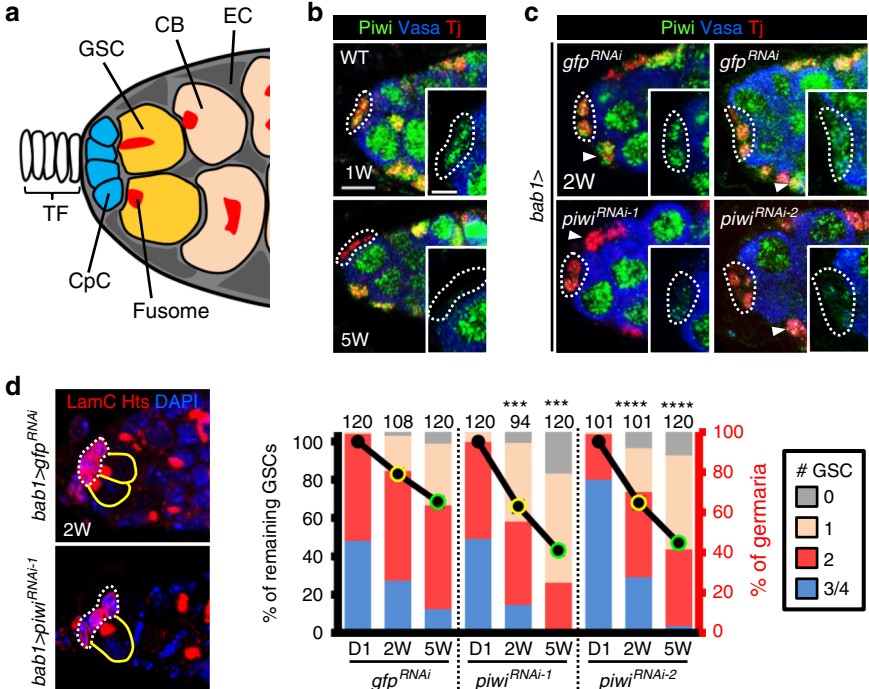

**Fig. 1 Aging reduces Piwi expression in the niche leading to GSC loss. a** The anterior region of *Drosophila* germarium. TF terminal filament, CpC cap cell, E escort cell, GSC germline stem cell, CB cystoblast. **b** Piwi expression is decreased in CpCs during aging (green, Piwi; blue, Vasa for germ cells; red, Tj for nuclei of CpCs and ECs). Scale bar, 5 μm. Scale bar in inset image, 2.5 μm. **c** Piwi expression is decreased in 2-week-old CpCs of *bab1 > piwi^{RNAi}* germaria (green, Piwi; blue, Vasa; red, Tj). Arrowheads, ECs. **d** Left, GSC number is decreased in 2-week-old *bab1 > piwi^{RNAi-1}* germaria (red, LamC for CpC nuclear envelopes; red, Hts for fusomes; blue, DAPI for DNA). Dashed white circles, CpCs; yellow circles, GSCs. Right, GSC number in *bab1 > gfp^{RNAi-1}*, *bab1 > piwi^{RNAi-1}*, and *bab1 > piwi^{RNAi-2}* germaria at 1 day (D1), 2 weeks (2W), and 5 weeks (5W) after eclosion. The numbers of analyzed germaria are shown above each bar. Percentage (%) of remaining GSCs (left *y*-axis) represents average GSC number at each time-point normalized to D1 within the same genotype; % of germaria (right *y*-axis) indicates the proportion of germaria carrying 0, 1, 2, 3, or 4 GSCs for the indicated genotype and age. Error bars of left *y*-axis, S.E.M., ***$P < 0.001$ (0.00011 for 2W *piwi^{RNAi-1}* and 0.0008 for 5W *piwi^{RNAi-1}*); ****$P < 0.0001$, yellow and green outlines on symbols indicate that the % of remaining GSCs is significantly lower than D0 of the same genotype. Green outline on symbols indicates that the % of remaining GSCs is significantly lower than symbols with yellow outline within same genotype. Student's *t*-test was used for comparisons of % remaining GSCs. All images (**b**, **c**, and **d**) are the same magnification and share same scale bar with **b**. All inset images (**b**, **c**) are the same magnification and share same scale bar with insert image of **b**. At least two biological replicates were performed for each experiment.

than control flies. This 32–34% loss of GSCs from D1 to 2 weeks in *bab1 > piwi^RNAi* flies was similar to the loss observed in control flies from D1 to 5 weeks (Fig. 1d), indicating that depletion of Piwi in the niche accelerated the GSC aging phenotype by 3 weeks. From these data, we conclude that Piwi expression in the young niche maintains GSC number.

**Niche Piwi silences transposons to maintain GSCs.** We next tested if Piwi silences transposons in the niche. We first monitored transposon activity in the germaria with *gypsy-lacZ*, a transcriptional reporter of the *gypsy* retrotransposon[20], which is among the most active endogenous insect retroviruses[21–23] and is known to be silenced by a Piwi-piRNA complex[20]. The *gypsy-lacZ* reporter was not expressed in the 1-week-old germarium but was strongly expressed in CpCs of 5-week-old flies, indicating derepression of transposons in the aged niche (Fig. 2a and Supplementary Fig. 1h, i). As expected, *gypsy-lacZ* expression was present in CpCs of *bab1 > piwi^RNAi-1* but not in control flies at 2 weeks after eclosion (Fig. 2b). Consistently, *gypsy* transcripts, examined by in situ hybridization, were also highly increased in 2-week-old *piwi*-KD CpCs (Supplementary Fig. 2). Interestingly, *gypsy* transcripts were also highly abundant in control and *piwi*-KD TFs (Supplementary Fig. 2), where Piwi was not expressed. We noticed that in 2-week-old *bab1 > piwi^RNAi-1* germaria, anterior ECs also showed reduced Piwi expression (see Fig. 1c, arrowhead) but did not express *gypsy-lacZ* (Fig. 2b). In addition, knocking down Piwi specifically in adult ECs did not cause GSC reduction (Supplementary Fig. 3 and Supplementary Table 1), supporting our previous conclusion that Piwi in CpCs maintains GSCs.

To examine which retrotransposons are suppressed by Piwi, we isolated CpCs (GFP-positive) from *bab1 > piwi^RNAi-1 gfp* and *bab1 > gfp* flies for RNA sequencing (Supplementary Fig. 4). Notably, TF cells were isolated with the CpCs, as they also expressed GFP. However, TF cells do not express Piwi, or only express it at very low levels[24] (see Supplementary Fig. 1e). Interestingly, a number of retrotransposons were already expressed in control niche cells (Fig. 2c). For instance, *copia* accounted for 27% of the total reads (164014 reads per kilobase per million mapped reads, RPKM; Supplementary Table 2). In *piwi*-KD niche cells, total retrotransposon transcripts were increased (214625 RPKM), with *copia* transcripts remaining very high (Fig. 2c and Supplementary Table 2). Increases in the expression levels of *ZAM* (47-fold), *mdg1* (10-fold), *blood* (10-fold) and *gypsy* (1.26-fold) were observed in *piwi*-KD compared to control niche cells (Fig. 2c). This moderate increase of *gypsy* transcripts in isolated *piwi*-KD niche cells could be due to the presence of TF cells, which expressed high levels of *gypsy* transcripts (See Supplementary Fig. 2). Therefore, it appears that Piwi preferentially silences a subset of retrotransposons in the niche.

To determine whether the increase of retrotransposons in *piwi*-KD CpCs caused GSC loss, we suppressed reverse transcription of replicating retrotransposons by feeding *bab1 > gfp^RNAi-1* and *bab1 > piwi^RNAi* flies with (−)-L-2′,3′-dideoxy-3′-thiacytidine (3TC), a cytidine analog that is clinically used to inhibit reverse transcription of human immunodeficiency virus and Hepatitis B virus for suppressing viral replication mediated by complementary DNA (cDNA) generation[25,26]. Consistent with our previous experiment, GSC maintenance in 2-week-old *bab1 > piwi^RNAi* flies was significantly decreased as compared to controls (Fig. 2d, e and Supplementary Table 1), although *piwi^RNAi-2* expression had a milder effect on GSC loss (26%) in this experiment (Fig. 2e and see Fig. 1d). Two-week-old control flies with or without 3TC treatment after eclosion showed comparable GSC numbers, while GSC loss in 2-week-old *bab1 > piwi^RNAi* flies was restored to the

control level upon feeding with 3TC for 2 weeks after eclosion (Fig. 2d, e and Supplementary Table 1).

To test the role of elevated retrotransposons in *piwi*-KD niche cells for GSC maintenance, we used the only two publicly available *RNAi* lines to suppress *gypsy* and *copia* in the *piwi*-KD niche. Strikingly, knocking down the retrotransposon *gypsy*, but not *copia*, in the *piwi*-KD niche restored GSC number to the control level (Fig. 2e, and Supplementary Table 1), suggesting the retrotransposon type-specific effect on GSC maintenance. Thus, Piwi-mediated silencing of retrotransposons in the niche, and inhibition of *gypsy* transposons in particular, serves to maintain GSCs. Interestingly, rescue by knockdown of *gypsy* retrotransposon was also found with regard to the neurodegeneration phenotype in a fly TDP-43 model[21,27]. The relative importance of this particular retrovirus may be due to the facts that the *gypsy* retrotransposon is highly active, encodes an endogenous retrovirus, and its replication can generate de novo insertions in aged fly brain[9]. Overall, we show that *gypsy* upregulation is at least largely responsible for the GSC-loss phenotype, but our results do not completely rule out roles for other retrotransposons on the GSC-loss phenotype.

**Niche Piwi supports GSC anchorage.** Previous studies reported that transposon jumping leads to cell death via DNA damage-induced Chk2/p53 signaling pathway[21,28–31]. Although approximately 30% of GSCs were lost in 2-week-old *piwi*-KD flies, CpC number was not affected (Supplementary Fig. 5a). In addition, DNA damage, as revealed by γ-H2Av[32], was observed in the CpCs of 30% of 5-week-old germaria ($n = 120$) accompanied by a reduction of CpC number (Supplementary Fig. 5a–c). However, removing one copy of *p53* or co-knockdown of *loki* (encoding fly Chk2) in the *piwi*-KD niche did not restore GSC numbers (Supplementary Fig. 5d and Supplementary Table 1). These results suggest that niche function, but not niche cell survival, is disrupted when retrotransposons are activated.

To maintain GSCs, CpCs provide both Dpp stemness signals[33] and physical contact via cell-cell adhesion, which is mediated by trans-interaction between extracellular domains of E-cadherin[34] and further strengthened by the cytoplasmic interaction of Cadherin-Catenin-Actin[35]. We did not observe decreased *dpp* mRNA production in the niche, nor did we find decreased Dpp signaling in the GSCs of *bab1 > piwi^RNAi* ovaries (Supplementary Fig. 6). Therefore, we next asked whether the *piwi*-KD niche exhibits defective GSC anchoring by examining E-cadherin, Armadillo (Arm, the β-catenin orthologue in fly) and filamentous (F)-actin levels. In *bab1 > piwi^RNAi* germaria, total expression of E-cadherin was the same as control in CpCs (Fig. 3a, f), however, its expression was enriched in CpC-CpC junctions but dramatically decreased in CpC-GSC junctions (Fig. 3a, g). In contrast, Arm expression in *bab1 > piwi^RNAi* germaria was reduced in CpCs (including CpC-GSC junctions) but remained relatively high in the junction between CpCs (Fig. 3b, h, i). This localization of Arm may explain the enrichment of E-cadherin in CpC-CpC junction of the *piwi*-KD niche. Arm reduction was also found in flies with CpCs carrying homozygous mutations of *piwi*[1] or *piwi*[2] (null and hypomorphic allele, respectively)[36], induced by mitotic recombination[37] (Supplementary Fig. 7). Consequently, F-actin expression was also decreased in the junction between *piwi*-KD CpCs and GSCs (Supplementary Fig. 8). We further showed that Arm expression in the niche was necessary for E-cadherin accumulation in the CpC-GSC junction for GSC anchorage (Supplementary Fig. 9a–c); during aging, Arm and F-actin expression levels were decreased in the CpC-GSC junction (Supplementary Fig. 9e–h), along with the decreased E-cadherin[13]. These results suggest that Arm affects E-cadherin expression and cellular localization. Thus, we conclude that Piwi

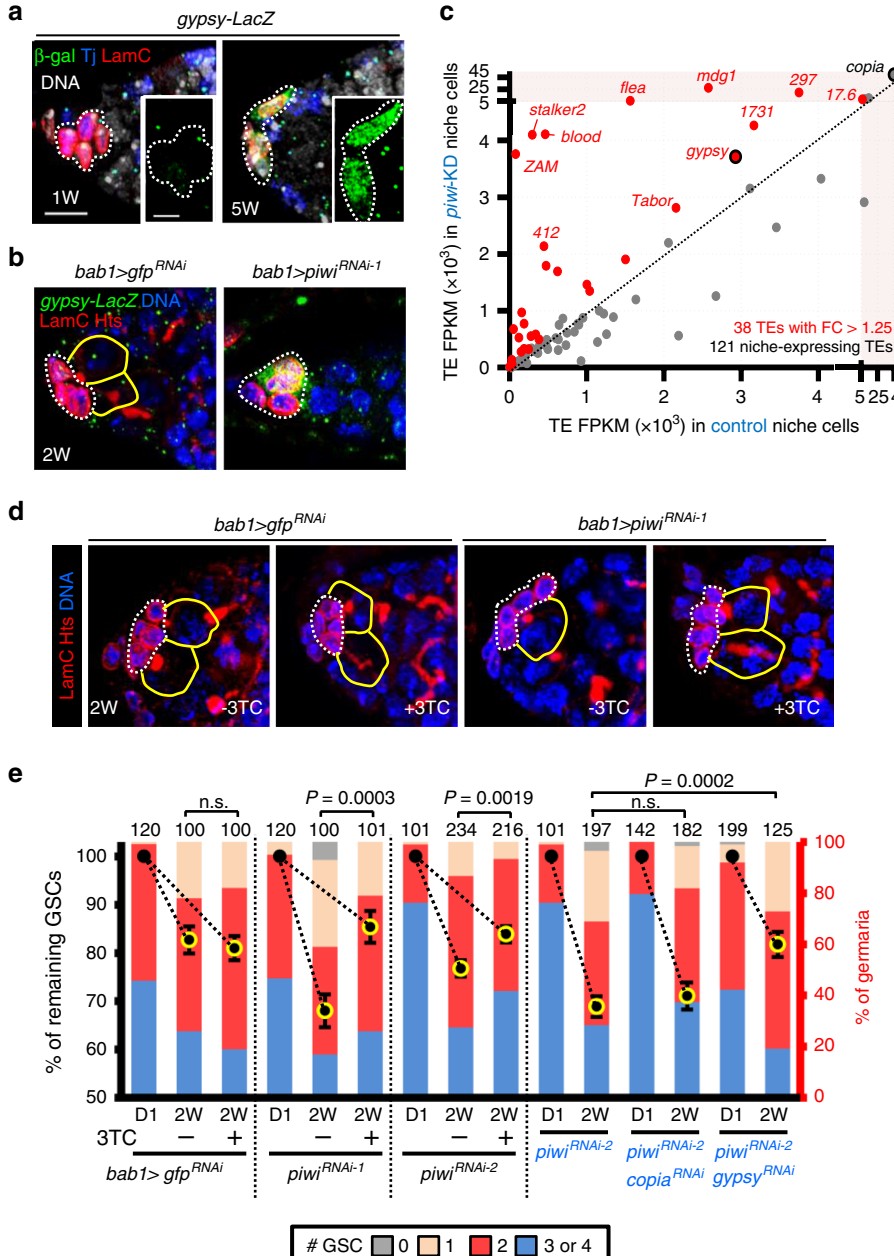

**Fig. 2 Piwi suppresses retrotransposons for GSC maintenance. a** *gypsy-lacZ* expression was increased in CpCs of 5-week (W)-old females (green, LacZ retrotransposon reporter; blue, Tj for CpC and EC nuclei; red, LamC for CpC nuclear envelopes; gray, DAPI for DNA). Scale bar, 5 µm. Scale bar in inset image, 2.5 µm. **b** *gypsy-lacZ* expression is increased in CpCs of the 2-week-old *bab1>piwi^RNAi-1* germarium (green, LacZ retrotransposon reporter; blue, Tj; red, LamC; red, Hts for fusomes). Dashed circles, CpCs; yellow circles, GSCs. **c** Retrotransposon expression profiles in sorted control and *piwi*-KD niche cells. **d** Two-week-old *bab1>gfp^RNAi-1* and *bab1>piwi^RNAi-1* germaria with or without 3TC treatment (10 mM) (Red, LamC and Hts; blue, DAPI). **e** GSC numbers in 2-week-old germaria of indicated genotypes with or without 3TC treatment. Percentage (%) of remaining GSCs at 2W (left y-axis) represents GSC number at 2W with or without 3TC treatment normalized to GSC number of the same genotype at D1 (newly eclosed) shown in Fig. 1d. Note that GSC number of *piwi^RNAi-2* alone at D1 in 7th and 10th columns is the same data shown in Fig. 1d. % of germaria (Right y-axis) indicates the proportion of 2-week-old germaria carrying 0, 1, 2, 3, or 4 GSCs. Error bars of left-y-axis in **e**, S.E.M. n.s., non-significant. Yellow outline on symbols indicates that the % of remaining GSCs is significantly lower than D0 of the same genotype. Student's t-test was used for comparisons of % remaining GSCs. All images (**a**, **b**, and **d**) are the same magnification and share scale bar with **a**. All inset images (**a**) are the same magnification. At least two biological replicates were performed for each experiment.

in the niche promotes Arm expression to support E-cadherin-mediated niche-GSC adhesion.

**Retrotransposons impair niche-GSC anchorage via GSK3.** Notably, *arm* mRNA expression was not decreased in *piwi*-KD CpCs (see Supplementary Fig. 3e), raising the possibility that Piwi

may control β-catenin stability via Glycogen synthase kinase 3 (GSK3), which phosphorylates β-catenin for degradation[38]. Indeed, suppressing the expression of Shaggy (Sgg, the GSK3 orthologue in fly) or overexpressing a constitutively active form of Arm (Arm^S10, which cannot be phosphorylated by GSK3[39,40]) in *piwi*-KD CpCs rescued Arm expression (Fig. 3d, h), E-cadherin

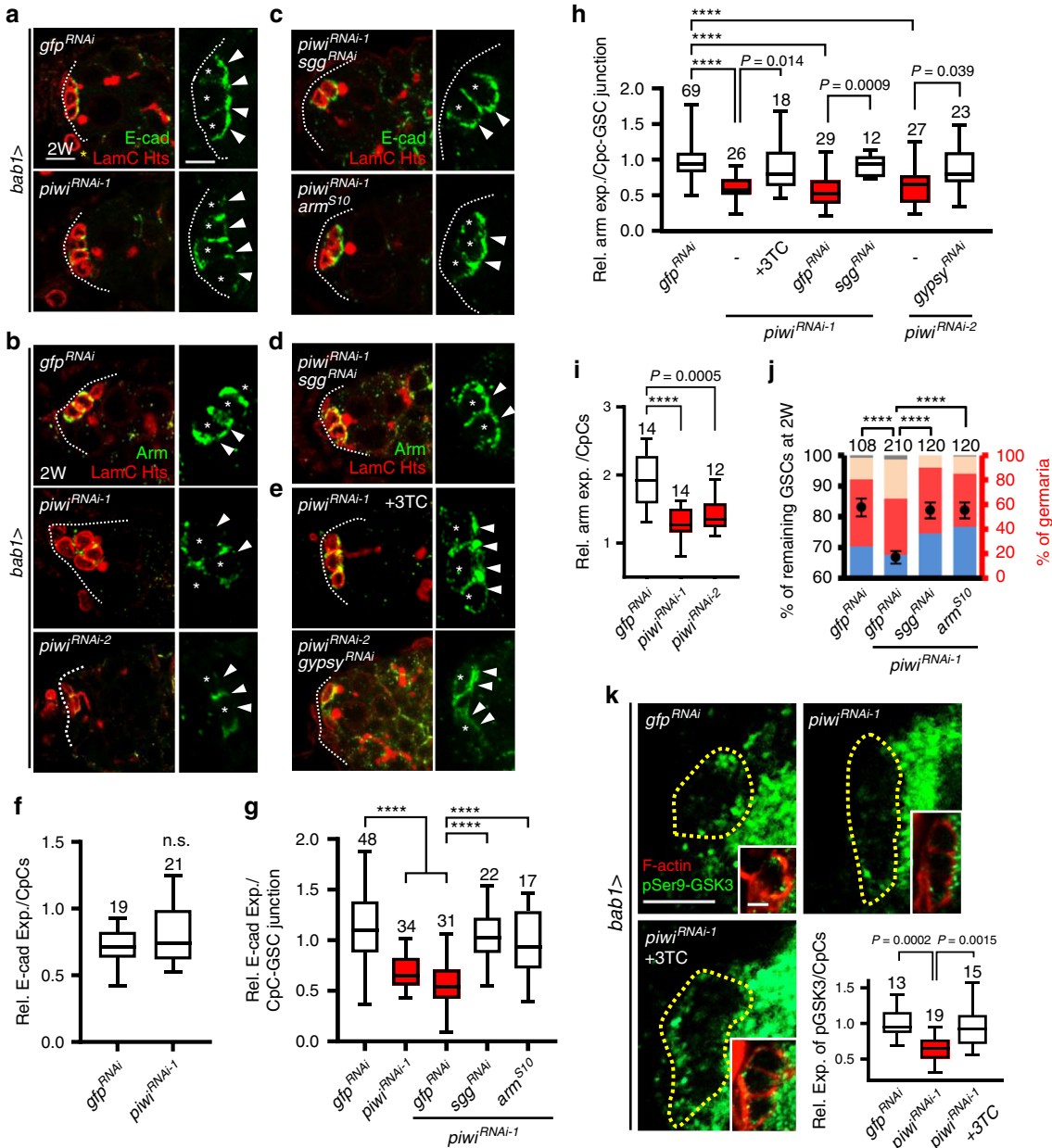

**Fig. 3 Retrotransposons increase GSK3 activity to impair niche-GSC anchorage. a, c** E-cadherin expression was evaluated at the niche-GSC junction (arrows) of 2-week (W)-old *bab1 > piwi^RNAi-1*, *bab1 > piwi^RNAi-1sgg^RNAi*, and *bab1 > piwi^RNAi-1arm^S10* germaria (green, E-cadherin; red, LamC for CpC nuclear envelopes; red, Hts for fusomes). Scale bars in left and right images are 5 and 2.5 μm, respectively. **b, d, e** Arm expression in the niche-GSC junction of 2-week-old *bab1 > piwi^RNAi-1*, *bab1 > piwi^RNAi-2*, *bab1 > piwi^RNAi-1;sgg^RNAi*, *bab1 > piwi^RNAi-2;gypsy*RNAi, and germaria of *bab1 > piwi^RNAi-1* with 10 mM 3TC treatment (green, Arm; red, LamC and Hts). Same magnification of images (**a–e**, left), and of enlarged images (**a–e**, right) share the left and right the scale bar in a, respectively. **f, g** Relative expression of E-cadherin in the niche (**f**) or niche-GSC junction (**g**) of germaria with indicated genotypes. **h, i** Relative expression of Arm in the niche-GSC junction (**h**) or niche (**i**) in the germarium of indicated genotypes. **j** Relative percentage (%) of GSC number in 2-week-old germaria of indicated genotypes (normalized to GSC number at D1). Note that GSC number of *gfp^RNAi* alone are the same data as shown in Fig. 1d. % of germaria (Right *y*-axis) indicates the proportion of 2-week-old germaria carrying 0, 1, 2, and 3 or 4 GSCs. **k** pSer9-GSK3β expression in the niche of 2-week-old *bab1 > gfp^RNAi* and *bab1 > piwi^RNAi-1* germaria with or without 10 mM 3TC treatment (green, pSer9-GSK3β; red, F-actin for CpC plasma membranes). Scale bars, 5 μm; all inset images in **k** are same magnification (scale bar, 2.5 μm). Dashed line, the anterior edge of the germarium (**a–e**); asterisks, CpCs; dashed circles outline niche CpCs (**k**). Box-Whiskers plots (**f, g, h, i**, and **k**) show the central lines for median values, edges of box represent upper (75th) and lower (25th) quartiles and whiskers show minimum and maximum values. Numbers of analyzed niches/niche-GSC junctions and GSCs are shown above or within each bar (**f–k**). Error bars, S.E.M. in **j**. Statistical analysis, Student's *t*-test for **f** and **j**; One-way ANOVA for **g, h, i**, and **k**. ****$P < 0.0001$. At least two biological replicates were performed for each experiment.

localization in the CpC-GSC junction (Fig. 3c, g), and GSC number (Fig. 3j). Further, 3TC treatment or co-knockdown of *gypsy* retrotransposon in *bab1 > piwi*[RNAi] flies also restored Arm expression in *piwi*-KD CpCs (Fig. 3e, h). These results suggest that retrotransposons cause Arm degradation via GSK3, which is highly regulated by its phosphorylation status[41].

Phosphorylation of GSK3 at Tyr216 (pTyr216) is an activating event, while Ser9 phosphorylation (pSer9) is inhibitory. In *piwi*-KD CpCs, expression of pSer9-GSK3 was reduced (Fig. 3k), but not pTyr216-GSK3 or total GSK3 (Supplementary Fig. 10), and this reduction was prevented by 3TC treatment (Fig. 3k). We also found that pSer9-GSK3 levels in CpCs were decreased in 7-week-old control flies (Supplementary Fig. 11a, b), and overexpressing a constitutively active form of GSK3 (Sgg[S9A]) in the niche reduced Arm and E-cadherin expression, as well as GSC number (Supplementary Fig. 11c–g). Therefore, Piwi in CpCs maintains expression of Arm by suppressing GSK3 activity via inhibitory phosphorylation.

**Piwi supplementation in the aged niche slows GSC loss.** Since constitutively overexpressing *piwi* in CpCs of adult flies delayed age-dependent GSC loss (Supplementary Fig. 12), we proceeded to ask if GSC loss mediated by niche Piwi deficiency can be reversed or slowed down by intervention within a specific time-window during aging and whether the Piwi-Arm regulatory axis is involved in age-dependent GSC loss. To address these questions, we supplied Piwi or Arm expression to the aged niche by utilizing a niche *GeneSwitch GAL4, 2261*[42], which can be activated by the steroid RU486 (mifepristone)[43,44]. Of note, this driver shows some basal leaky activity in the absence of RU486[42,45]. Overexpressing either Piwi or Arm[S10] in 3- or 6-week-old niches by feeding flies with RU486 for 1-week delayed GSC loss (Supplementary Fig. 13a–f and Supplementary Table 1). Consistent with these results, overexpressing Piwi in the 6-week-old niche for 1 week also partially rescued Arm expression and E-cadherin distribution in the CpC-GSC junction, although total E-cadherin expression levels remained low (Supplementary Fig. 13g–l). These results suggested that Piwi-independent regulation of E-cadherin expression levels and GSC loss occurs during aging, and together, the findings reflect the critical involvement of transposon activation in the aged niche.

**Toll-GSK3 signaling in Piwi-deficient niche causes GSC loss.** Replicated retrotransposons may integrate into new loci within the genome and affect expression of genes that mediate inhibitory phosphorylation of GSK3. To test whether this occurs, we isolated GFP-positive *piwi*-KD niche cells (CpCs, anterior ECs and TFs) and other ovarian non-niche somatic cells (GFP-negative) from 2-week-old *bab1 > gfp & piwi*[RNAi-1] germaria for whole genome sequencing (Supplementary Fig. 14a). We found 70 new transposon insertion sites in the genome of the *piwi*-KD niche compared to the control genome; 53 insertion events were associated with 22 different retrotransposons (Supplementary Table 3). However, none of these new insertions were located in genes known to regulate inhibitory phosphorylation of GSK3. Surprisingly, we did not find any new insertions of the *gypsy* retrotransposon. We also did not identify any de novo *gypsy* transposition events in the aged CpCs using the *gypsy-TRAP* reporter system[9], which contains Ovo binding hotspots for *gypsy* insertion (Supplementary Fig. 14b–d). These results suggest that GSK3 activation in the *piwi*-KD niche is likely mediated by a retrotransposition (mediated by retrotransposon cDNA)-independent mechanism.

Like the *Drosophila gypsy* retrotransposon, the human retrotransposon Human endogenous retrovirus K (HERV-K) can

generate endogenous virus[46], which is present in some patients with amyotrophic lateral sclerosis[47]. HERV is known to induce immune response via Toll-like receptor (TLR) signaling[48,49], which activates GSK3 by reducing inhibitory phosphorylation, leading to β-catenin degradation in enterocytes of mice[50]. It is therefore possible that retrotransposons may induce Toll-mediated signaling in the *piwi*-KD niche to activate GSK3 and subsequently cause GSC loss. Therefore, we next asked if Toll signaling is involved in the transposon-GSK3 axis that impairs GSC maintenance. In the fly, Toll and Toll-7 signaling may be activated by virus infection[51,52]. Moreover, Toll-5 is known to interact with Toll and is involved in Toll signaling activation[53]. We found that depletion of Toll or Toll-5, but not Toll-7, in *piwi*-KD CpCs rescued GSC loss (Fig. 4a, b and Supplementary Table 1) and knocking down Toll or Toll-5 alone in the CpCs did not cause increased maintenance of GSCs (Supplementary Table 1). Decreased levels of pSer9-GSK3 and Arm, and mislocalization of E-cadherin in the *piwi*-KD CpCs were also rescued by depletion of Toll or *gypsy* retrotransposon (Fig. 4b, c and Supplementary Fig. 15).

**Niche Piwi inhibits Viral-like particles and Toll signaling.** Upon activation of Toll-mediated immune signaling, Cactus (orthologue of mammalian IκB[54]) is phosphorylated and degraded in the cytoplasm to release Dorsal (orthologue of mammalian NF-κB[54]). Dorsal then enters the nucleus, where it activates transcription of genes encoding antimicrobial peptides (AMPs)[55]. To test whether this immune signaling pathway is activated in *piwi*-KD CpCs, we examined expression of Cactus and Dorsal. We found that Cactus and Dorsal were highly expressed in the cytoplasm of *bab1 > gfp*[RNAi] CpCs (Fig. 4d), while both Cactus and Dorsal expressions were dramatically reduced in the cytoplasm of aged CpCs, *piwi*-KD CpCs, and CpCs that overexpressed a constitutively active form of Toll, [Toll[10b]][56] (Fig. 4d). Notably, we did not observe nuclear location of Dorsal, nor did we find increases of known AMPs in isolated *piwi*-KD CpCs by transcriptome analysis. This lack of AMP signals may have been due to amplification bias in the cDNA from isolated CpCs, our examination of unregulated AMPs, or quick removal of Dorsal from the nucleus. Furthermore, knockdown of *gypsy* or *toll* in the *piwi*-KD niche or 3TC treatment of *bab1 > piwi*[RNAi-1] flies did not reverse Cactus degradation (Fig. 4f), suggesting that other Toll receptors may be involved in the activation of immune signaling, and this process may be independent of the effects of *gypsy* and retrotransposition. More interestingly, we observed the presence of viral-like particles (VLPs) by transmission electron microscopy in the cytoplasm of many *piwi*-KD (12/37 = 32% of germaria from ten ovaries) and aged CpCs (8/36 = 22% of germaria from ten ovaries), but VLPs were not observed in control CpCs (0/22 germaria from 10 ovaries) (Fig. 5a). Meanwhile, 3TC treatment reduced the number of VLPs in *piwi*-KD CpCs (Fig. 5b, c), suggesting that retrotransposon replication can increase transposon transcripts for VLP assembly. However, VLPs are not reduced in the *piwi/gypsy*-double KD CpCs (Fig. 5a); this result is consistent with our finding that *gypsy* did not undergo retrotransposition and suggests that VLPs may be derived from non-*gypsy* retrotransposons.

According to the results of our genetic and 3TC treatment experiments (Fig. 6a), we predicted that activation of Toll signaling in the aged or *piwi*-KD niches would both promote GSK3 activity and degrade Cactus. However, we found that knockdown of *gypsy*, or inhibition of reverse transcription for virus cDNA generation in the *piwi*-KD niche suppressed GSK3 activity but did not restore Cactus levels nor did it completely clear VLPs. We therefore speculate that retrotransposons other

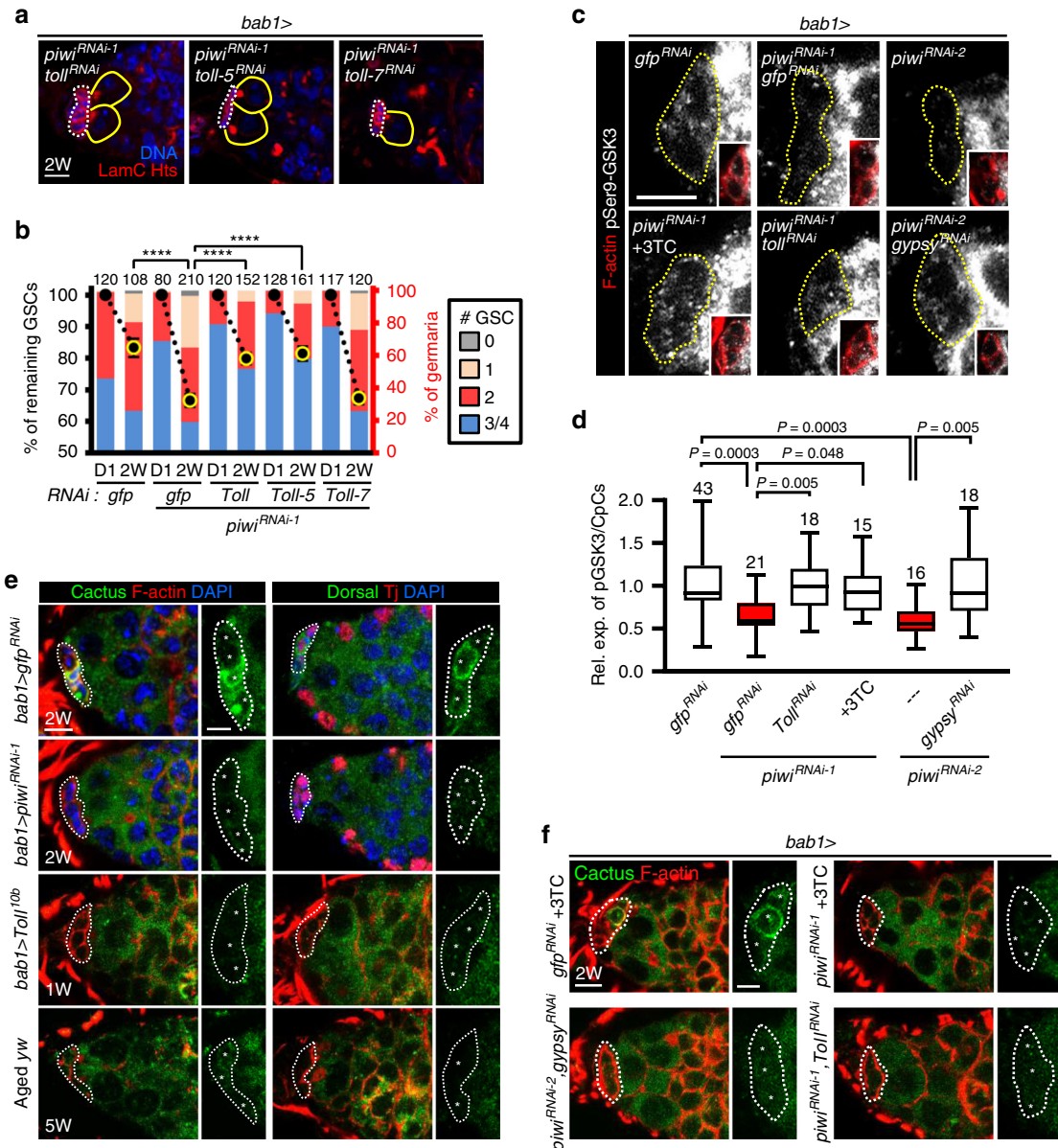

**Fig. 4 Decrease of Piwi expression in the niche causes GSC loss via Toll-mediated GSK3 activation. a**, **b** GSC number in 2-week-old germaria of indicated genotypes. Percentage (%) of remaining GSCs at 2W (left y-axis) represents GSC number at 2W normalized to GSC number at D1 of the same genotype. Note that GSC number of *gfp*[RNAi-1] alone and *gfp*[RNAi-2]; *piwi*[RNAi-1] are the same data as shown in Fig. 1d and Fig. 3j, respectively; % of germaria. Right y-axis indicates the proportion of 2-week-old germaria carrying 0, 1, 2, and 3 or 4 GSCs. Yellow outline on symbols indicates that the % of remaining GSCs is significantly lower than D0 of the same genotype. Error bars, S.E.M. **c**, **d** pSer9-GSK3β expression in the niche of 2-week-old *bab1* > *piwi*[RNAi-1], *bab1* > *piwi*[RNAi-2], *bab1* > *piwi*[RNAi-1] & *toll*[RNAi], *bab1* > *piwi*[RNAi-2] & *gypsy*[RNAi] and *bab1* > *piwi*[RNAi-1] with 3TC treatment (10 mM) germaria (white, pSer9-GSK3β; red, F-actin for CpC plasma membranes). Box-whiskers plots (**d**) show the central lines for median values, edges of box represent upper (75th) and lower (25th) quartiles and whiskers show minimum and maximum values. **e** Cactus (left panel: Cactus, green; red, F-actin; blu**e**, DAPI) and Dorsal (right panel: Dorsal, green; red, Tj; blue, DAPI) expression in *gfp*-KD, *toll*[10b]-expressing, *piwi*-KD, and aged CpCs of flies at indicated age. **f** Cactus (green; red, F-actin) expression in 3TC treatment of *gfp*-KD or *piwi*-KD, and double knockdown *gypsy* or *Toll* in CpCs of flies at indicated age. Scale bars, 5 μm for images on left and 2.5 μm for enlarged images on right. Student's t-test for **b** and One-way ANOVA for **d**. ****P < 0.0001. At least two **b**iological replicates were performed for each experiment.

than *gypsy* may contribute to VLP generation and Cactus degradation. Furthermore, co-knockdown of *piwi* and *toll* in the niche cannot prevent Cactus degradation, suggesting that other Toll receptors may be involved in Toll-Cactus signaling. However, we cannot rule out the possibility of Piwi-dependent, Toll-independent Cactus expression. Nevertheless, our results suggest that in the aged niche, decreased Piwi expression results in generation of retrotransposon-derived viral materials (such as viral transcript, cDNA, or protein) and coincident activation of

Toll signaling, which acts through GSK3 to degrade Arm (Fig. 6b). This reduction in Arm disrupts Cadherin-Catenin-Actin adhesion, compromising GSC maintenance.

## Discussion

It is possible that this pathway for removal of GSCs from the aged or *piwi*-KD niche would protect the integrity (avoid retrotransposon integration) of genomic material that is transmitted to

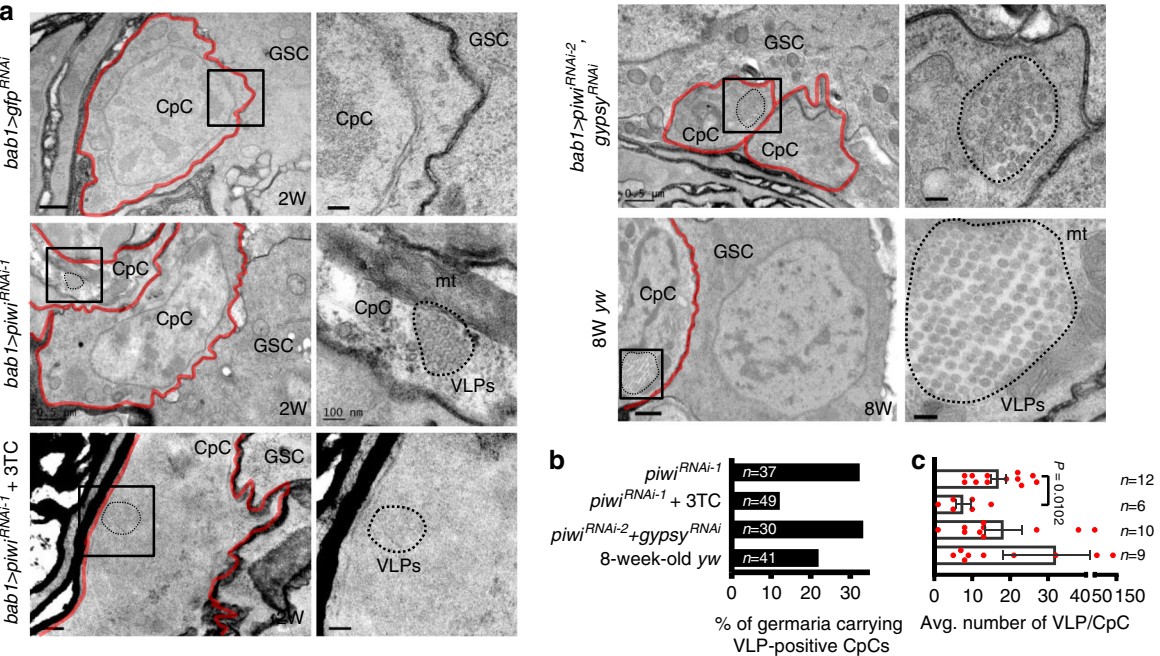

**Fig. 5 Retrotransposons generate virus-like particles in Piwi-deficient CpCs. a** TEM images of anterior germaria of indicated genotype, age and treatment. Right images are enlargements of boxed areas in left images. CpCs red lines, VLPs virus-like particles (black dashed circle), mt mitochondrion. Scale bars, 0.5 μm for left images, except left image of *bab1* > *piwi*^RNAi-1 with 3TC treatment, in which scale bar is 200 nm. Scale bars, 100 nm for all right images. **b** % of germaria containing VLP-positive CpCs in indicated genotypes. 3TC treatment, but not co-knockdown *gypsy*, reduced percentage of VLP-positive *piwi*-KD niche-containing germaria. Examined germaria numbers (*n*) from biologically independent samples are showed inside the bar. **c** Quantification of VLPs in TEM sample of indicated genotypes. 3TC treatment reduced VLP number in VLP-positive CpCs. The number of germaria containing VLP-positive CpC from biologically independent samples are showed at the right position of dot plot. Error bars, S.E.M (**c**). Statistical analysis, Student's *t*-test for **c**. At least two biological replicates were performed for each experiment.

the next generation. Interestingly, reduced expression of Piwi and Arm with coincident derepression of the *copia* retrotransposon were also observed in aged hub cells within the male GSC niche (Supplementary Fig. 16). Hence, Piwi may play a similar role in the GSC niche of both sexes, but the individual retrotransposons that are suppressed by Piwi may differ between males and females.

Intriguingly, flies carrying a mutation of *piwi* lacking a nuclear localization signal (Piwi^NT) were reported to display elevated *gypsy* expression and normal GSC number[57], which appears to contradict our findings. However, *gypsy* expression in *piwi*^NT mutant ovaries is only half as abundant as it is in *piwi*^2 mutant ovaries[57]. In addition, *piwi* null mutant ovaries are extremely small, while *piwi*^NT mutant ovaries display normal morphology and carry egg chambers, each with a monolayer of follicle cells, where *gypsy* is expressed when Piwi is deleted[58]. Thus, the reported elevation of *gypsy* is likely to partly rely on expression by follicle cells, which are far away from GSCs. In addition, Piwi protein contains a Piwi domain, which has been shown to cleave RNA in vitro[59]. Based on this activity, cytoplasmic Piwi might possibly participate in degrading retrotransposons, in addition to its well-characterized role in suppressing retrotransposon transcription.

It is not clear how retrotransposons, retrovirus or virus materials (including transcripts, cDNA made by reverse transcription and/or proteins) in the cytoplasm are detected by Toll receptors in CpCs. Using the *Drosophila* RNAi Screening Center Integrative Ortholog Prediction Tool (DIOPT)[60], we found that Toll and Toll-5 are the respective orthologues of human TLR7/9 and TLR4 (Supplementary Table 4), which sense viral materials[61]. TRL7/9 are present and active in endosomes within the cytoplasm to sense viral materials[61]; similarly, endosomal location of Toll

has been reported, and its activation requires the endocytic pathway[62], suggesting that Toll may also participate in virus sensing. It is not clear if Toll-5 senses viral materials. Nevertheless, it is known to dimerize with Toll, and it is required for Toll-mediated immune activation[53]. Another open question is why aging only reduces Piwi expression in the GSC niche. Given that many types of mammalian stem cell niches lose their function during aging[63,64], it would be interesting to know if transposon activation may broadly account for this decline.

Inflammation is associated with aging-related disease[65]. A very recent study showed that aging-induced elevation of LINE1 retrotransposons triggers inflammation in mice, and 3TC treatment reduces this inflammation[66]. Together with this previous study, our results may shed light on the long-known association among retrotransposons, inflammation/innate immunity, and Alzheimer's disease (AD)[67,68]. The pathogenesis of AD is known to involve increased GSK3 activity as a result of Aβ42 accumulation[69], although the underlying mechanisms are still not clear. Our study raises the possibility that retrotransposon-immune signaling-GSK3 regulation observed in the fly GSC-niche unit may also occur in mammalian neural cells. Indeed, GSK3 activity in an AD model in human embryonic kidney 293 cells was suppressed by 3TC treatment (Supplementary Fig. 17). Moreover, in an AD fly model with human Aβ42 overexpression in the eye, GSK3 activity was suppressed and neural activity was partially rescued by 3TC treatment (Supplementary Fig. 18). In addition to AD, several other aging-related diseases were reported to exhibit both derepressed retrotransposons and elevated GSK3 activity, including multiple sclerosis[70,71], amyotrophic lateral sclerosis[72,73], and cancer[74,75]. Given that GSK3 plays a crucial role in inflammation[76], our results not only show that retrotransposon derepression impairs stem cell maintenance via

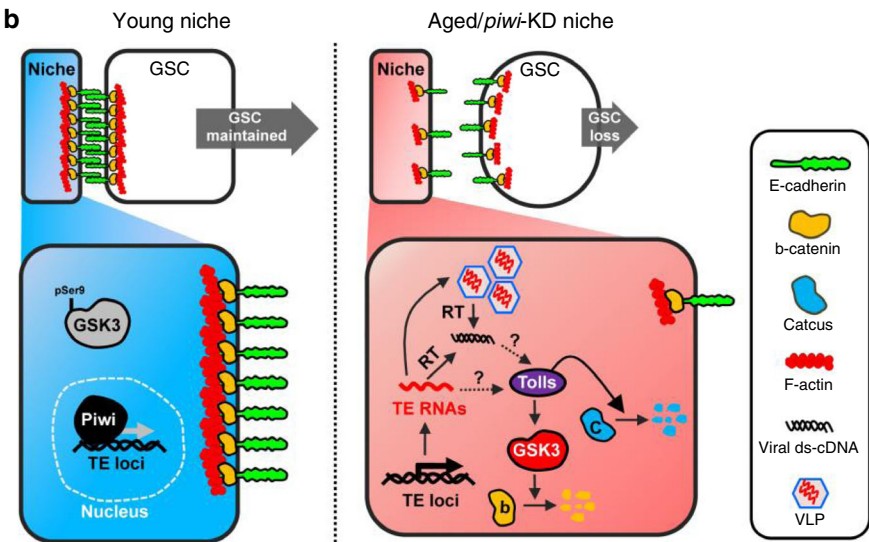

**Fig. 6 Proposed model of how disrupted Piwi-mediated retrotransposon silencing impairs GSC maintenance. a** Summary of phenotypes after aging, genetic manipulation and 3TC treatment of flies with *piwi*-KD niches. Y indicates presence of a given phenotype; R indicates rescue of the given phenotype; PR indicates partial rescue of the given phenotype; n.d. indicates not determined. **b** Model of how retrotransposon silencing by Piwi in the niche serves to maintain GSC attachment. Aging derepresses transposons/transposable elements (TEs) via reduction of Piwi. Decreased Piwi level derepresses TEs, which generate viral materials and activate Toll-GSK3 signaling to promote degradation of β-catenin. Loss of β-catenin impairs Cadherin-Catenin-Actin-mediated cell adhesion, leading to GSC loss. The mechanisms by which retrotransposons activate Toll receptor will require further examination. Dashed arrows with question marks indicate proposed effects.

Toll-GSK3 signaling but also highlight the possibility that retrotransposon reactivation may contribute to aging-related tissue dysfunction and diseases via immune signaling-mediated GSK3 activation.

## Methods

**Fly stocks and culture**. *Drosophila* stocks were maintained at 22–25 °C on standard medium (normal diet). *yw, w^1118, Canton S*, and *Oregon* were used as wild-type controls. Null *p53^-ns* (a gift from Dr. John M. Abrams, University of Texas Southwestern Medical Center, U.S.A.) alleles have been described previously[57,77,78]. *myc-piwi*, a genomic construct with a Myc-tag inserted between Piwi amino acid residues 3 and 4, was used to monitor Piwi expression[16]. *gypsy-lacZ* (a *lacZ* reporter fused with the promoter and 5′UTR of *gypsy*; Bloomington Drosophila Stock Center [BDSC] #53723) and *bamp-Bam:GFP* (*bam:gfp* coding sequences driven by the *bam* promoter; a gift from Dr. Michael Buszczak, University of Texas Southwestern, U.S.A.) were used to monitor *gypsy* and *bam* transcription[20,79–81], respectively. UAS-RNAi or transgenic lines used in this study are listed as follows: UAS-gfp^RANi-1 (BDSC #9331)[82], UAS-gfp^RNAi-2 (BDSC #9330)[83], UAS-piwi^RNAi-1 (Vienna Drosophila Resource Center [VDRC] #101658)[24], UAS-piwi^RNAi-2 (BDSC #33724)[84], UAS-arm^RNAi-1(VDRC #7767)[85], UAS-arm^RNAi-2 (VDRC #107344)[86], UAS-loki^RNAi (VDRC #44980)[21], UAS-sgg^RNAi-1 (BDSC #31308)[87], UAS-sgg^RNAi-2 (BDSC #38293)[88], UAS-toll^RNAi (BDSC #35628), UAS-toll-7^RNAi (BDSC #30488), UAS-toll-5(tehao)^RNAi (BDSC #17903), GMR-GAL4 (BDSC #9146), UAS-Abeta-m26a (BDSC #33769), UAS-Sgg^S9A (BDSC #5255)[89], UAS-Toll^10b (BDSC #58987), UAS-copia^RNAi (a gift from Dr. Peng Jin, Emory University School of Medicine, U.S.A.), UAS-gypsy^RNAi, gypsy-TRAP and mutated gypsy-TRAP[9] (gifts from Dr. Joshua Dubnau, Stony Brook University School of Medicine, U.S.A.), UAS-arm^S10 (a constitutively active form of Armadillo; BDSC #4782)[39,40], UAS-HA-Piwi (a gift from Dr. Eric Lai, Memorial Sloan Kettering Cancer Center, U.S.A.)[84]. The *bab1-GAL4, ptc-GAL4* and *c587-GAL4* lines have been previously described[90,91]. *2261-GAL4^GeneSwitch (GS)* is a driver that activates expression of *UAS* transgenes in TF and CpCs upon the binding of RU486 (mifepristone) to the progesterone receptor-GAL4 chimera[42,44]. Other genetic elements are described in Flybase (http://flybase.org). Flies expressing *RNAi* or transgenes driven by *bab1-GAL4* or *ptc-GAL4* also carried a temperature sensitive (ts) mutant GAL80 under the control of a tubulin promoter (*tub-GAL80^ts*) and were cultured at 18 °C to suppress GAL4 activity during developmental stages; the flies were switched to 29 °C after eclosion to degrade GAL80^ts and activate GAL4. For experiments, 10–15 females plus three wild-type male flies were maintained in a plastic vial and fed with normal diet plus dry yeast powder. For RNAi knockdown experiments, food (normal food with dry yeast powder) was changed every other day. For the experiments involving co-knockdown of *piwi* and *gypsy*, *gypsy^RNAi* and *piwi^RNAi-2* were used together, as these transgenes are respectively located on the second and third chromosomes, allowing for efficient of the presence of both *RNAi* lines.

**Genetic mosaic analysis.** Genetic mosaic clones were produced by Flipase (FLP)/ FLP recognition target (FRT)-mediated mitotic recombination[37]. Female of [*neoFRT40A/Cyo*], [*FRT40A, piwi^1/CyO*] or [*FRT40A, piwi^2/CyO*] (gifts from Dr. Haifan Lin, Yale school of medicine, USA) were crossed with male of [*hsflp; FRT40A, ubi-GFP/CyO*], respectively, to obtain female larva with [*hsflp/+; neoFRT40A/ FRT40A*] (as control), [*hsflp/+; FRT40A, piwi^1/FRT40A, ubi-GFP*] and [*hsflp/+; FRT40A, piwi^2/FRT40A, ubi-GFP*]. To generate CpC clones, mid-third instar larva (mid-L3) stage of female larva were subjected to heat shock for 1 h at 37 °C, twice a day for three days. After heat shock, females raised at 25 °C were transferred to fresh food daily for 2 weeks until dissection.

**Pharmacological treatments**. Flies were cultured at 25 °C before RU486 treatment and maintained as described above; at the desired age, flies were switched to vials containing normal diet plus a wet yeast paste mixed with RU486 (M8046, Sigma). A 10 mg/mL stock solution of RU486 (mifepristone; Sigma) was made in pure ethanol (Sigma). For RU486-containing wet yeast preparation, 0.28 mL of RU486 stock solution, 0.1% blue food color additive (to confirm food intake; #861146, Sigma) and 1.52 mL ddH$_2$O were mixed well and then added to 1 g active dry yeast (RED STAR) for a final concentration of 3.6 mM RU486.

For 3TC (2′, 3′-Dideoxy-3′-thiacytidine, #L1295, Sigma) treatment, newly eclosed flies were cultured in plastic vials containing normal diet plus a 3TC-containing wet yeast paste. For 3TC-containing wet yeast preparation, 11.5 mg 3TC, 0.1% blue food color and 3.3 mL ddH$_2$O was mixed well and then added to 1.7 g active dry yeast for a final concentration of 10 mM 3TC. Control flies were fed with the same diet but without RU486 or 3TC. Food was changed daily until flies were dissected.

**Immunostaining and fluorescence microscopy**. Ovaries were dissected in Grace Insect Medium (GIM) (#11605094, Thermo Fisher) and then fixed in 5% (v/v) paraformaldehyde (#43368, Alfa Aesar)/Grace's Insect Medium for 13 min at room temperature (RT). Ovaries were washed with 0.1% PBST (0.1% Triton-X100) and then teased apart in 0.1% PBST and incubated in blocking solution (PG0033, GOALBIO) for 3 h at RT, and with primary antibodies for 16 h at 4 °C. Testes were dissected in PBS and then fixed in 4% (v/v) paraformaldehyde in PBS for 20 min at RT. Testes were washed by 0.3% PBST for 5 min and incubated with 0.3% NaDoC/ 0.1 or 0.3% PBST for 30 min. After washing with 0.3% PBST, testes were incubated with blocking solution (3% BSA/0.3% PBST) for 30 min at RT, and with primary antibodies in 0.3% BSA (in 0.3% PBST) for 16 h at 4 °C. Eyes were dissected in PBS and then fixed in 4% paraformaldehyde (#15710, Electron Microscopy Sciences) for 45 min at RT, followed by 3 washes of 15 min each in 1% and then 3 washes in 0.4% PBST (PBS + Triton X100). The samples were then incubated with primary antibodies overnight at 4 °C. After washing well in PBST, ovaries, testes or eyes were then incubated with corresponding secondary antibody at 4 °C for 16 h. Primary antibodies included: mouse anti-Hts antibody (1:30; 1B1, Developmental Studies Hybridoma Bank (DSHB)), mouse anti-Lamin (Lam) C antibody (1:25; LC28.26, DSHB), mouse anti-Cactus antibody (1:500; 3H12, DSHB), mouse anti-Dorsal antibody (1:100; 7A4, DSHB), Rat anti-E-cad antibody (1:3; ECAD-2, DHSB), mouse anti-Arm antibody (1:4; N27A1, DHSB), rabbit anti-pMad antibody (1:200; #1880, Epitomics), mouse anti-β-gal antibody (1:500; Promega), mouse anti-Piwi antibody (1:1000; a gift from Dr. Mikiko C. Siomi, University of Tokyo, Japan), guinea pig anti-Traffic jam (Tj) antibody (1:5000; a gift from Dr. Dorothea Godt, University of Toronto, Canada), rabbit anti-Histone H2AvD pS137antibody (1:1000; #600-401-914, Rockland), mouse anti-GSK3α/β (1:200; MA3-038, Thermo Fisher Scientific), anti-pSer9-GSK3β (1:200; #5558, Cell Signaling), anti-pTyr216-GSK3 (1:150, #05−−413, Millipore), and rabbit anti-GFP (1:1000; Torrey Pines). Mouse anti-ENV antibody (1:100, a gift from Dr. Joshua Dubnau, Stony Brook University School of Medicine, U.S.A.; ENV protein signals were not observed in *piwi*-KD niche but can be observed in *flamingo* mutants with reduced piRNAs against *gypsy* retrotransposon [data not shown]). AlexaFluor 488, 568 or 633-conjugated goat anti-mouse, anti-rabbit or anti-rat secondary antibodies (1:500; Molecular Probes) were used as appropriate. For F-actin staining, samples were stained with Rhodamine- or Alexa Fluor™ 488-Phalloidin (1:80, R415 and 1:200, #A12379, respectively, Thermo Fisher Scientific). Samples were stained with 0.5 mg/mL DAPI (Sigma) and kept in mounting solution (80% glycerol containing 20 mg/mL N-propyl gallate, Sigma) at −20 °C until mounting; eye samples were mounted in Vectashield (H-1000 and H-1200, Vector Laboratory). Images were collected on a Zeiss LSM 700 confocal microscope at 40x or 63x magnification with Zen software (2010 version) and modified in LSM image browser (version 4,2,0,121).

**Image analysis and quantification**. GSCs were identified by the presence of a fusome (labeled with anti-Hts antibody), which is an intracellular structure adjacent to CpCs. CpCs contain ovoid nuclear envelopes that were stained by LamC. Rhabodomere numbers in each Ommatidium were analyzed by Fiji. Regions of interest for analyzing protein expression were outlined manually. To analyze the levels of Piwi, *gypsy-lacZ*, pSer9-GSK3, pTyr216-GSK3, total GSK3 and pMad in germarial cells, Image J (National Institutes of Health [NIH], U.S.A.) was used to measure the mean fluorescent intensity (arbitrary units) of the confocal z-section with the highest fluorescent signal from each cell type. The protein levels were normalized to the indicated proteins (used as a staining control) in the same cell type, or in other germarial cells (Supplementary Fig. 19). Tj in the same cell type was used for normalization of Piwi and *gypsy-lacZ* in CpCs, and Piwi in ECs. Expression of Piwi in GSCs was normalized to Vasa expression in GSCs. For pSer9-GSK3, pTyr216-GSK3 and total GSK3 levels in CpCs, the mean fluorescent intensity was normalized to the level in TFs. In RU486-treatment experiments, Piwi expression in CpCs was normalized to Piwi expression in follicle cells. To analyze the total amounts of E-cadherin and Arm expressed in the niche, confocal z-stacks of CpCs were merged. The mean fluorescence intensity of the merged image was measured with ImageJ (version 1.50i) and normalized to the mean fluorescence intensity for E-cadherin or Arm in a germ cell of a 16-cell cyst (which is derived from a CB) in the same germarium. To analyze E-cadherin, Arm or F-actin

expression in the niche-GSC junction, the confocal z-section with the highest signal of E-cadherin, Arm or F-actin in the niche-GSC junction was analyzed and normalized to the signal for E-cadherin or Arm in a germ cell in a 16-cell cyst, or the signal for F-actin at the junction between CpCs of the same germarium. To increase sample size, 20 flies of each genotype were collected for dissection. Only a few germaria from each ovary were examined, such that the total number of examined germaria reached about 100 for GSC number measurements and 10 for expression intensity measurements. To assess the distribution of E-cadherin at the interface between GSCs and CpCs, three-dimensional reconstructions were made from confocal z-sections using Imaris (version x64, 9.1.2) (Bitplane).

**Statistics and reproducibility**. Quantitative data were recorded and graphed in Excel (Microsoft) or GraphPad Prism 8 software (version 8.3.0). P-values were calculated from two-tailed unpaired Student's t-test or chi-squared test in Excel for comparison between two groups, and One-way ANOVA in Prism (GraphPad) for multiple comparisons. P-values are showed as exact value or asterisks in the graphs and legends of figures. At least two biological replicates were performed for each experiment. Representative images were taken from at least two technical replicate experiments with similar results or from independent biological samples with similar phenotypes.

**RNA in situ hybridization**. RNA in situ hybridization was performed as previously described[92]. In brief, ovaries were dissected and fixed overnight with 4% paraformaldehyde in PBS (1% DMSO and 0.1% DEPC) at 4 °C. Ovaries were washed with PBST (PBS with 0.1% Tween-20), dehydrated by a series of ethanol solutions with increasing concentration (25%, 50%, 75%, and 100%) and kept at −20 °C for at least 16 h. Ovaries were then rehydrated by a series of ethanol solutions with decreasing concentration and treated with proteinase K (50 μg/mL in PBST; Sigma) for 5 min at RT, followed by post-fixing in 4% paraformaldehyde for 30 min. The tissues were prehybridized in hybridization solution (50% formamide, 5x SSC, 0.1% Tween-20, 50 μg/μL heparin, 100 μg/mL yeast t-RNA and 10 μg/mL salmon sperm DNA) at 60 °C for 1 h, then hybridized with hybridization solution containing Digoxigenin (DIG)-labeled probes at 60 °C overnight. Hybridized ovaries were washed with 50 and 25% hybridization solution diluted in 2x SSC, PBST, and then incubated with 3% H$_2$O$_2$ in PBST to inactivate endogenous perioxidase (POD). Samples were then blocked with 2x blocking buffer in maleic acid (Roche) at RT for 1 h, then incubated with anti-DIG-POD (1: 500, Roche # 11207733910) in blocking buffer at 4 °C overnight. After washing, signals were developed using a TSA plus fluorescence kit (Perkin Elmer).

For the preparation of *dpp* RNA probes, the coding region (nucleotides 846 to 2403) of *dpp* transcript was amplified from adult fly ovary cDNA (described below) using *dpp* primers (5′-AGGACGATCTGGATCTAGATCGGT-3′ and 5′-ACTTTGGTCGTTGAGATAGAGCAT-3′)[86]. For *gypsy* RNA probes, the region of *gypsy* retrotransposon from nucleotides 6000 to 7020 was amplified from adult fly ovary cDNA (described below) using *gypsy* primers (5′-CATCAATAAGGTGATCAATGC-3′ and 5′-TGCTACGAAGCAATACATTG-3′). The fragments were subcloned into the pGEM-T Easy vector (Promega). Antisense RNA probes labeled with digoxigenin-UTP (Roche) were synthesized from 1 μg of NcoI-digested PGEM-T-piwi plasmid using the ampliCapTM SP6 high-yield message marker kit (Cell Script).

**Cell culture and Aβ enzyme-linked immunosorbent assay (ELISA) analysis**. Human embryonic kidney (HEK) 293-derived CG cells, which carry a tetracycline-inducible C-terminally Gal4/VP16-tagged the 99 amino acid C-terminal fragment of Amyloid Precursor Protein (APP-C99) and a Gal4 promoter driven firefly luciferase reporter gene[93], were seeded onto a 6-well plate ($2 \times 10^5$/well) and cultured for 24 h at 37 °C. Each well contained a final volume of 3 mL of Dulbecco's Modified Eagle Medium (DMEM, Sigma) plus 10 % Fetal bovine serum (FBS), 5 μg/mL blasticidin, 250 μg/mL zeocin, and 200 μg/mL hygromycin B. After 24 h, medium was replaced by serum-free DMEM containing Tetracyclin (1 μg/mL) with or without 3TC (10 μM) for 72 h. APP-C99 is cleaved by γ-secretase to form Aβ peptides, including Aβ40. Cells were then harvested for Western blot analysis (see Western blotting section below), and cell media was collected for Aβ ELISA analysis to detect secreted Aβ40 peptides using human Aβ40 ELISA kit (#KHB3481, Thermo Fisher Scientific) according to the manufacturer's instructions.

**Western blotting**. For the fly study, six pairs of fly eyes per group were dissected and homogenized in 2× Laemmli sample buffer (126 mM Tris/Cl, pH 6.8, 20% glycerol, 4% SDS and 0.02% bromophenol blue) containing 10% β-mercaptoethanol, after which the samples were boiled for 15 min. Protein lysates were collected from supernatant after centrifugation at 4 °C, 13,000 rpm for 15 min. For the cell line study, cells were lysed in RIPA buffer (20 mM Tris-HCl, pH 7.5, 150 mM NaCl, 1 mM EDTA, 1% NP-40, 1 mM PMSF and 50 mM NaF) with 2× protease inhibitor cocktail (Roche) for 30 min on ice with vortex every 5 min, and then were centrifuged at 4 °C, 13,000 rpm for 15 min. Supernatant was collected as protein lysate, which was then mixed with an equal volume of 2× Laemmli sample buffer and boiled for 15 min for further analysis. Proteins were separated on 10% SDS polyacrylamide gels (SDS-PAGE) and blotted onto polyvinylidene difluoride (PVDF) membranes. Membranes were blocked by 5% milk protein in TBST (0.1%

Tween-20) for 1 h at RT, then incubated with primary antibodies in TBST containing 5% milk protein at 4 °C for 16 h. After washing, membranes were incubated with secondary antibodies in TBST containing 5% milk protein at RT for 1 h. Signals were detected by chemiluminescence with a Western LightningTM Plus-ECL kit (PerkinElmer). Rabbit anti-phospho-Ser9 GSK3 (Cell Signaling Technology, #5558, 1:4000), mouse anti-GSK3α/β (1:200; sc-7291, Santa Cruz) and rabbit anti-Histone H3 (Abcam, #ab1791, 1:2000) were used as primary antibodies; horseradish peroxidase (HRP)-conjugated goat anti-rabbit IgG (Jackson ImmunoResearch, 1:10,000) was used as secondary antibody.

**CpC sorting for transcriptome analysis**. The transparent portions of 60 of 2-week-old *bab1 > gfp* and *bab1 > gfp & piwi^RNAi-1* ovaries (switched to 29 °C after eclosion) were dissected in GIM with 10% FBS. Ovaries were further dissociated in 2.5% Trypsin solution (Sigma) containing 600 units (U) Collagenase type I (Thermo Fisher Scientific) for 30 min at 29 °C with vigorous shaking; throughout the incubation, samples were vortexed every 5 min. Cell suspensions were filtered twice through a 70-μm nylon mesh. Cells were collected by centrifugation at $2000 \times g$ for 5 min at 4 °C, then re-suspended in GIM containing 10% FBS. Dead cells were labeled with propidium iodide (PI) (2 μg/mL). Three hundred GFP-positive cells (CpCs, TFs and few anterior ECs) from *bab1 > gfp* and *bab1 > gfp & piwi^RNAi-1* were collected into wells of 96-well plate containing lysis buffer from SMART-Seq HT kit (Takara Bio USA, Inc.) by fluorescence-activated cell sorting (FACS) on a FACSAriaII (BD Biosciences) (Supplementary Fig. 20a). Two biological replicates were used for the analysis. Total RNA was subjected to reverse transcription for cDNA generation in a 96-well plate, according to the manufacturer's instructions for the SMART-Seq HT kit. cDNA was further amplified by PCR (11 cycles), also according to manufacturer's protocol for the SMART-Seq HT kit. Amplified cDNA was purified by Agencourt AMPure XP (Beckman Coulter, Inc.). All RNA-seq protocols followed the manufacturer's instructions (Illumina). cDNA was qualified with a Bioanalyzer 2100 on a RNA 6000 labchip (Agilent Technology), and library construction was carried out with an Agilent SureSelect Strand Specific RNA library Preparation Kit for 75SE (Paired-End) sequencing on a Solexa platform. The sequences were determined by sequencing-by-synthesis technology with a TruSeq SBS kit. Sequencing data (FASTQ reads) were generated using the Welgene Biotech pipeline based on the Illumina base-calling program, bcl2fastq v2.20, at 14.5 M to 18.9 M (million reads) per sample. The sequences were then filtered to obtain qualified reads. Sequence quality trimming was performed with Trimmomatic v0.36, using a sliding-window approach. The transposon and gene expression levels were calculated as FPKM (Fragment Per Kilobase of transcript per Million mapped reads). For transposon analysis, the "Transposable elements (canonical set)" database was downloaded from FlyBase and used as a reference for transposon annotations. For coding-gene expression analysis, the reference gene annotations were retrieved from FlyBase.

**CpC sorting and genomic sequencing and analysis**. The transparent portions of hundreds of 2-week-old *bab1 > gfp & piwi^RNAi-1* ovaries (switched to 29 °C after eclosion) were dissected in GIM with 10% FBS. Ovaries were further dissociated in 2.5% Trypsin solution (Sigma) containing 600 units (U) Collagenase type I (Thermo Fisher Scientific) for 30 min at 29 °C with vigorous shaking; throughout the incubation, samples were vortexed every 5 min. Cell suspensions were filtered twice through a 70-μm nylon mesh. Cells were collected by centrifugation at $2000 \times g$ for 5 min at 4 °C, then re-suspended in GIM containing 10% FBS. Dead cells were labeled with propidium iodide (PI) (2 μg/mL). GFP-positive cells (piwi-KD CpCs, TFs—which do not express Piwi—and a few anterior ECs) and GFP-negative ovarian somatic cells (germ cells were excluded by their large size) were collected by fluorescence-activated cell sorting (FACS) on a FACSAriaII (BD Biosciences) (Supplementary Fig. 20b).

Genomic DNA was extracted from GFP-positive niche cells (CpCs and TF) and GFP-negative ovarian non-niche somatic cells (like ECs and FCs, which served as the *piwi*-KD control), as well as 50 ovaries from newly eclosed *bab1 > gfp & piwi^RNAi-1* of flies (cultured at 18 °C to silence GAL4 activity for the genotype control). Genomic DNA of whole ovaries was extracted by phenol-chloroform extraction, according to the instructions from Vienna Drosophila Resource Center (VDRC). Genomic DNA extracted from sorted GFP-positive niche cells and GFP-negative ovarian non-niche somatic cells (~600 cells for each sample) was amplified using the REPLI-g Single Cell Kit (Qiagen) according to the manufacturer's instructions. In brief, isolated cells were lysed by adding Buffer D2 and incubating on ice for 10 min. After adding stop solution, amplification master mix containing Phi DNA polymerase was added to the cell lysate and incubated at 30 °C for 2.5 h. Amplified genomic DNA was then purified using a DNeasy Blood and Tissue Kit (Qiagen) with EB elution buffer. Purified genomic DNA of isolated cells and of whole ovaries was subjected to genomic DNA sequencing by HiSeq Rapid v2 (paired-end 2 × 150 base pair). Transposon insertion sites were detected by Transposon Insertion and Deletion AnaLyzer (TIDAL)[94] based on the reference genome BDGP Release 6 (dm6). The software packages and annotation files were automated, and file format conversion, read trimming, mapping, and polymorphic transposon detection were performed seamlessly. Transposon insertion events were categorized by the location and types of the inserted sequences. New TE insertions found in GFP-positive niche cells but not in GFP-negative cells or ovary samples were called.

**Transmission electronic microscopy (TEM)**. TEM was performed similar to a previous study[95], with minor modifications. Fly ovaries were immersed in fixation solution (2.5% glutaraldehyde, 1% tannic acid and 2% paraformaldehyde in 0.1 M sodium cacodylate buffer) overnight at 4 °C. Tissues were then washed in 0.1 M cacodylate buffer with 0.2 M sucrose and 0.1% $CaCl_2$ three times at 4 °C (10 min for each wash). Next, ovaries were fixed with 1% osmium tetroxide, $OsO_4$ in 0.1 M cacodylate buffer for 1 h at 4 °C and washed three times with $ddH_2O$ at 4 °C (5 min for each wash). Ovaries were then stained with uranyl acetate at 4 °C for 1 h and washed with $ddH_2O$ three times at 4 °C (5 min for each wash). Stained samples were dehydrated with increasing concentrations of ethanol (30%, 50%, 75%, 90% and 3 × 100%, 5 min each) at 4 °C and infiltrated by 100% ethanol: Spurr = 3:1, 1:1, 1:3 (each for 1 h) and pure Spurr four times (1 h each time). Spurr was allowed to polymerize for 48 h at 60 °C. Ultrathin sections were sectioned with a diamond knife (DiATOME) at on a microtome and stained with 4% uranyl acetate for 20 min. After rinsing with H2O six times, ovaries were immersed in Reynolds lead citrate for 10 min. Then, slices were rinsed with H2O and mounted to on copper slot grids and observed under TEM. After mounting, germarial sections were examined by with a Tecnai G2 spirit TWIN transmission electron microscope (FEI Company) equipped with a Multiscan Gatan camera (Gatan) at an accelerating voltage of 120 kV.

**Electroretinogram (ERG)**. Electroretinograms (ERGs) were performed by immobilizing flies after 2- or 10-day light exposure with PVA glue on glass slides. The recording and reference electrodes were filled with 2 M NaCl while the ERGs were carried out. An iWorx 404 multichannel recorder was used to amplify the electrode voltage (iWorx Systems Inc., NH, UAS), and signal was filtered through an intracellular microelectrode amplifier (Walter Instrument Co., CT, USA). Light stimuli (1 s) were administered by a computer-controlled white LED system, and the ERGs were recorded using iWorx LabScribe2 software.

**Reporting summary**. Further information on research design is available in the Nature Research Reporting Summary linked to this article.

## Data availability

Transposon expression profiles (Raw FPKM) are provided in Supplementary Table 3 of Supplementary Information section. The source data underlying all quantification of GSC numbers and protein expression and uncropped immunoblots for each figures and supplementary figures are provided as Source Data file. All other data supporting the findings of this study are available from the responding author upon reasonable request. Source data are provided with this paper.

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

## Acknowledgements

We thank J.T. Dubnau, A.A. Aravin, V.A. Gvozdev, M.C. Siomi, D. Godt, H. Lin, P. Jin, the Bloomington Drosophila Stock Center, NIG-FLY, the VDRC Stock Center, and the DSHB for *Drosophila* stocks and antibodies. We also thank the Taiwan fly core for ordering fly lines and reagents, core facilities in the Institute of Molecular Biology, the Institute of Biomedical Science and the Institute of Cellular and Organismic Biology, as well as the NGS High-Throughput Genomics core at the Biodiversity Research Center, Academia Sinica for assistance with cell isolation, image analysis and genomic sequencing, Marcus Calkins for English editing; Chung, B.-C., Huang, F., and Tsai, M,-T. for valuable comments on this article. This work was supported by a thematic grant of Academia Sinica.

## Author contributions

K.-Y.L., W.-D.W., and H.-J.H. designed and interpreted the experiments and wrote the paper. C.-H.L. and K.-Y.L. contributed to image analysis of TEM. Y.-H.S. contributed to Supplementary Fig. 1. E.R., Y.-C.C., K.-Y.L., and H. P. contributed to testis staining. Y.-T.C., K.-Y.L. and Y.-F.L. contributed to analysis of Alzheimer disease in cell lines, and F.-Y.T., T.-Y.S., K.-Y.L., and C.-C.C. contributed to analysis of Alzheimer disease in flies. M.-Y.L., B.-Y.Y., S.-H.C. and C.-Y.L. helped whole-genomic sequence and data analysis for novel transposon transposition, respectively. K.-Y.L. performed the remaining experiments.

## Competing interests

The authors declare no competing interests.
