## [Peer Review File · Nature Communications]

Reviewers' comments:

Reviewer #1 (Remarks to the Author):

The manuscript of Lin et al. addresses the molecular mechanisms responsible for the decline in oogenesis observed in older flies. They reported that older flies demonstrate a loss of Germline Stem Cells (GSC) in the ovary. The loss of GSC in aged flies correlates with reduced expression of Piwi protein in cap cells that form GSC niche. Next, they used a temporally controlled approach to knock-down Piwi in cap cells and showed that it leads to loss of GSCs. Piwi loss in aged cap cells leads to de-repression of gypsy retrotransposon, consistent with prior studies that demonstrate the role of Piwi in transposon silencing. Interestingly, knock-down of gypsy expression using RNAi or administration inhibitor of reverse transcriptase was sufficient to rescue GSC number upon Piwi depletion. They went ahead to build connections between activation of gypsy (and other transposons) and GSC loss. This was done by first showing that depletion of Piwi weakens GSC anchorage on cap cells via E-cad/Arm/F-actin axis. Authors suggest that immune signaling detects viral-like particles originating from retrotransposons to change E-cad/Arm/F-actin pathway. Overall, results reported here could potentially interest a broad audience. However, there are several major concerns that need to be addressed.

Major concerns:

1. The link between toll signaling and retrotransposon de-repression requires further support. The authors showed that anti-RT drug reduces the number of viral-like particles. However, the link between viral-like particles and toll pathway is not based on experiments in the paper and instead relies on previous observations in mammals (citations 42-44). More work is needed to demonstrate causation relationship between expression of retrotransposons and toll signaling in *Drosophila*. For example, can over-expression of gypsy activate toll signaling? Also, it is not obvious how a drug that targets reverse transcriptase can reduce the number of viral particles.
2. The claim that Piwi plays an important role during aging is compromised by the inability to rescue the phenotypes by Piwi supplementation (Supp.Fig. 9). It is unclear why authors resorted to a sophisticated drug-induced over-expression of Piwi. Can constitutively overexpressing piwi by *bab1>piwi* prevent/slow aging? It is critical to demonstrate that if Piwi is not lost in cap cells during the 5-week period, aging phenotypes can be circumvented.
3. It is unclear why a single retrotransposon, gypsy, plays a major role in aging. According to author results, specific knock-down of gypsy expression is sufficient to rescue the loss of GSCs caused by Piwi depletion. This result indicates that other transposons do not play a significant role. However, other elements such as *mdg1* are upregulated more strongly (210-fold) upon Piwi depletion in OSC cells compared to gypsy. Given that knock-down of gypsy can rescue the phenotype to the same degree as the application of an anti-RT drug that supposedly reduces the activity of all retrotransposons, what is special about gypsy that is different from all other transposons? How much do other transposons (retro- and DNA transposons) contribute?
4. The cytoplasmic role of Piwi remains unknown, so it is difficult to interpret results from Piwi [NT] mutant that localizes exclusively in the cytoplasm or to explain how it maintains GSC. It has not been shown that cytoplasmic Piwi targets transposon transcripts for degradation as authors claimed (line 95, 98). The endonuclease activity of Piwi is not required for any function in vivo (Darricarrère et al., PNAS 2013). Without elucidating the molecular function of cytoplasmic Piwi, authors cannot comment on how cytoplasmic Piwi maintains GSC or silence retrotransposons.

Minor concerns:

1. Authors should describe the usage of Gal80ts and temperature shift in the main text to explain why KD starts after eclosion (line 57) and why studying *bab1>piwiRNAi* in 2W.
2. Phenotypes of all double KD will be more convincing if the control is co-KD an unrelated gene like *gfp*, instead of KD *piwi* alone where 2x as much drives goes to *piwiRNAi*.
3. It helps if authors can describe levels of F-actin, toll signaling and pSer9-GSK9 in aged cap cells, in addition to the KD model, to argue that they occur during aging.
4. It would make the manuscript much stronger if authors do RNA-seq in sorted niche cells. OSC can serve as a reference, but it describes other cell types like follicular cells.
5. Can there be aging-dependent, *piwi*-independent changes in E-cad/Arm/F-actin, GSK3 phosphorylation state, toll signaling? Complexity of aging could be discussed more.
6. Can 3TC prevent RT in GSCs as well to at least in part explain the rescue?
7. AD data in the end seem largely out of place. Adding these weakens the manuscript.
8. In Fig S9e, 7W 2261 without induction does not have aging-related GSC loss (~90% GSCs left). GSC loss during aging does not appear robust when compared with Fig 1d.
9. "niche" and "cap cells" are used somewhat interchangeably throughout the text. Authors should be more accurate when describing effects seen in cap cells only.
10. Line 178, will EC (in addition to TF+CpC) also be GFP-positive?
11. Method of retrotransposon analysis is poorly documented. For instance, which annotation is used? e.g. *springer* cannot be found in either RepBase or FlyBase.
12. Fig 2c, pie chart is an ineffective way to visualize data, where one cannot appreciate a de-repression of transposon without reading the numbers.
13. Nearly all "enlarged" images lack scales. Also, in Fig S7b, outline in enlarged image seems highly inconsistent with the outline in the original image.
14. In Fig 3b bottom row, it is not obvious that there are more than one LamC+ CpC. A more representative image should be shown.
15. While cap cells survive by 2W, there is statistically significant increase of DNA damage (Fig S4), which should be described in text.
16. Image segmentation and quantification are not sufficiently documented. e.g. are cells outlined manually? Is it reliable to use nuclear marks (*tj*, *piwi*, *lamC*) to outline cell boundary? For analysis at cell-cell boundary that is 3D, is 2D quantification reliable?

Reviewer #2 (Remarks to the Author):

In *Drosophila* females, the identity of the adult germ line stem cells (GSCs) is maintained by the ovarian somatic niche cells the so called cap cells (CpCs). CpCs exert their maintenance function by two ways: 1) They tightly bound GSCs with the cell-cell adhesion molecules beta-catenin and E-cadherin and 2) prevent the differentiation of GSC-s by providing a TGF-beta-type signal molecule. The absence of either the physical contact or the TGF-beta signalisation result in differentiation, i.e. loss of the GSC-s. It is well documented that number GSC is decline with age of females. The manuscript gives a novel mechanistic explanation of the role of CpCs in the age dependent GSC loss.

The authors describe that *Piwi* expression is dramatically decrease in aged CpC-s resulting de-repression of Gypsy retrotransposons. By making use of CpC specific *piwiRNAi* they were able to mimic the age dependent GSC loss phenotype. The authors proved that the transposon de-repression neither cause CpC death nor impair with the TGF-beta signalling. On the other hand, they show that the expression of beta-catenin decreases in CpCs and E-Cadherin disappears form GsC-CpC junctions therefore the physical connection between the CpC-s and GSC-s is compromised in aged, as well as in the *piwiRNAi* niches.

The manuscript unambiguously proves that the de-repression of Gypsy retrotransposons in the CpC-s cause the GSC loss phenotype: 1) Virus-like particles were found in *piwiRNAi* germaria, 2) a reverse transcriptase inhibitor (3TC), as well as the *gypsyRNAi* suppresses the *piwiRNAi* caused GSC loss phenotype.

During a carefully planned series of experiments the authors proved that the *Piwi RNAi* acts

through the Glycogen synthase kinase-(shaggy in *Drosophila*)-beta-catenin- (armadillo in *Drosophila*) E-Catherin pathway. The authors also proved that the Toll mediated immune signalling is activated in piwiRNAi niches, since mislocalization of E-cadherin in the piwiRNAi niche is rescued by Toll depletion.

In summary, the authors describe a novel "quality assurance system" for *Drosophila* adult female germ line stem cells. As symptom of aging, the Piwi transposon regulation system weakens in the stem cell niche. As a consequence, retrotransposons are deregulated and produce virus-like particles capable of infecting the nearby stem cells. De-repression of retrotransposons, however induces a Toll-mediated immune response which activates Glycogen synthase kinase. Glycogen synthase kinase then induces degradation of beta-catenin, which finally results in delocalisation of E-Catherine from CpC-GSC junction. The end result is elimination of GSCs from the stem cell niche infected with the virus-like particles.

The focus of the manuscript is the *Drosophila* female stem cell niche. However, the authors present a series of experiments demonstrating that the immune response to retrotransposon de-repression may be a general protective phenomenon. They showed that 1) Piwi and Armadillo expressions are reduced while expression of copia-lacZ is increased in the aged *Drosophila* male GSC niche. 2) 3TC treatment suppresses GSK3 activity in human cell model of Alzheimer's disease. 3) Using the fly Alzheimer model they proved that 3TC treatment partially suppresses GSK3 activity induced in the Ab42-overexpressing fly eyes. In summary, the results presented in the manuscript may be of interest to a wide range of readers

Critical comments:

1) The authors observed, and quoted the literature, that number of GSCs decreased with age. They also showed that piwiRNAi in CpCs was resulting in a decrease in GSC number. However, two weeks were needed for the complete loss of Piwi expression (line 57). The two weeks lag of the effect of RNAi could well be the result of the well-known protein perdurance. In line 61, the authors wrote: "As previously reported^{12,13}, the number of GSCs in control flies decreased with age (fig. 1d and Supplementary Table 1). This decrease was accelerated in two independent piwi-knockdown (KD) lines (fig. 1d and 64 Supplementary Table 1)." The authors then consistently, but I think wrongly, classify piwi silenced phenotype as an age dependent loss." In my opinion, if RNAi had an immediate effect on Piwi protein depletion, the authors would not write about an increase of age-dependent loss. It would be better to apply the term age-dependent loss only to the phenotypes measured in the aged wild type flies.

2) The authors clearly demonstrated that the piwi mediated protection from the harmful effect of Gypsy retrotransposon is strictly cap cell specific. The other two types of somatic niche cells, the terminal filament cells and the anterior escort cells are not involved in this process. TF cells do not express Piwi at all. Piwi expression is only slightly reduced in the very old, 8-week-old escort cells. To examine which types of retrotransposons are suppressed by Piwi in the niche cells, the authors examined a publicly available transcriptome database for retrotransposon expression levels in a piwi-knock down ovarian somatic cell line (OSC). In line 85 the authors wrote ".....ovarian somatic cells (OSCs), which resemble CpCs to support GSCs²¹." According to the citation no 21: „We assume that the somatic component of the cultures (from which the OSC cell lines were sub cloned) is derived from the somatic stem cells (cf. Fig. 1b)". This means that the origin of the OSC cell line is ambiguous and the used cell lines most probably do not represent CpCs properly. Therefore, I suggest to omit the in silico analysis of OSC expression patterns from the manuscript.

3) line 94 "Piwi acts in the nucleus through epigenetic mechanisms to suppress transposon transcription²², while cytoplasmic Piwi targets transposon transcripts for degradation²³. Ovaries carrying mutant Piwi with a nuclear localization defect (piwiNT) did not accelerate age-dependent GSC loss (Supplementary Fig. 3 and Supplementary Table1), suggesting that Piwi suppresses cytoplasmic retrotransposon transcripts for replication acting as the second defence mechanism to maintain GSCs."

Firstly, it seems to me that ref. 23 was wrongly cited. It is a general though in the field, that Piwi exclusively participates in co-transcriptional silencing that happens in the nucleus and does not participate in the cytoplasmic silencing. It is also generally accepted that in the ovarian somatic

cells only the co-transcription silencing but not the cytoplasmic silencing is functional. Secondly, The cited ref. 23 says: "Transcript level of the Gypsy and mdg1 retrotransposons increased 10- to 20-fold in the ovaries of homozygous piwiNt and piwi2 or transheterozygous piwi2/piwiNt flies relative to the corresponding heterozygotes (Fig. 3A)." Later: "Here we describe the phenotype of a unique mutation in the piwi gene that leads to the formation of cytoplasmic PiwiNt (i.e., N-truncated Piwi protein) lacking the NLS. The properties of this mutant made the direct influence of the piRNA pathway on GSC maintenance unlikely, as the piwiNt mutant displayed normal GSC self-renewal (Fig. 1 C and D) but lost Piwi mediated transposon repression completely in ovarian cells (Fig. 3A) including niche cells responsible for GSC self-renewal signalling (Fig. 4B). " There is a plain contradiction between the ref. 23 and the manuscript. According ref. 23, there is an elevated Gypsy level in piwiNt mutant germlaria (most probably in young ovaries since the ref. 23 did not focus on the ageing). According the manuscript, however, "Ovaries carrying mutant Piwi with a nuclear localization defect (piwiNt) did not accelerate age-dependent GSC loss." With the other words the elevated Gypsy level in the GSC niche might not be the reason of the age dependent GSC loss. This contradiction must be resolved in every way.

4) line 98: ".....suggesting that Piwi suppresses cytoplasmic retrotransposon transcripts for replication acting as the second defence mechanism to maintain GSCs. To test this hypothesis, we suppressed reverse transcription of replicating retrotransposons by feeding *bab1>gfpRNAi-1* and *bab1>piwiRNAi* flies with (-)-L-2',3'-dideoxy-3'-thiacytidine (3TC), a cytidine analogue that is clinically used to inhibit reverse transcription of human immunodeficiency virus and Hepatitis B virus^{24,25}.

If the cytoplasmic function of Piwi in transposon silencing had been true, the logic of the introduction of 3TC experiments would have been still incorrect. Namely, irrespective of whether the retrotransposon de-repression occurs in the nucleus or in the cytoplasm, 3TC experiment may give positive result. That is, by 3TC experiments it is not possible to test the hypothesis about the cytoplasmic activity of Piwi.

5) line 40: "Despite their genomic abundance, transposons are effectively silenced by Piwi-interacting RNAs (piRNAs) at transcriptional and post-transcriptional levels^{6,7}. However, this process is known to be attenuated in aged tissues⁸⁻¹⁰, where stem cells are frequently lost¹¹". In this context, "this process" refers to the piwi pathway. The authors refer three publications (No 8, 9, and 10) concerning the age-related attenuation of the Piwi system. Unfortunately, neither of the cited publications is about the Piwi pathway.

Reviewer #3 (Remarks to the Author):

This manuscript examines the age-dependent loss of germline stem cells in the *Drosophila* ovary. The authors combine molecular, imaging, and genomic data to propose a link between piwi and GSC loss. Specifically, they interpret the data to suggest that "retrotransposons in the aged GSC niche generate endogenous virus to activate Toll-mediated immune signaling, which subsequently activates Glycogen synthase kinase 3 (GSK3) to impair b-catenin-E-cadherin-mediated GSC anchorage, finally resulting in GSC loss". This is a very interesting hypothesis that would likely be of interest to a range of researchers studying stem cell biology, ageing, and niche function.

While I appreciate the amount of work presented in this study, and the high technical quality of the experiments, I have several concerns about claims contained in the manuscript. I have detailed my major concerns below.

1. Virus – Toll: In several instances, the manuscript discusses the generation of virus within the stem cell niche. These claims are not supported by the data. Unless the authors show that infectious cell-free virus particles capable of generating an integrated provirus in newly infected cells are produced, they cannot make any statements about virus production. The instances of virus-like particles in some TEMs are certainly interesting, but more work is required. Equally

importantly, claims of virus-dependent activation of Toll signaling require more support. To support their claims that the niche is activating a Toll-dependent antiviral response, the authors cite two publications that link Toll signaling to antiviral defenses in *Drosophila*. However, it is important to note that one of those studies established a role for Toll-7 in antiviral defenses. Importantly, the authors effectively exclude Toll-7 as a receptor. The second cited manuscript implicated Toll in fly antiviral responses. However, the Toll receptor is generally considered to recognize microbial patterns at the plasma membrane. In the context of this manuscript, it is not clear how Toll can detect cytosolic viral particles or genomes, nor do the authors present any data to indicate that is happening. It is possible that the putative antiviral responses under investigation here are being mediated by Toll-5, but there are no data to support an antiviral role for Toll-5, or to suggest that Toll-5 is able to bind viral particles or genomes in the cytosol. Furthermore, there are no data to support activation of the Toll pathway (e.g. phosphorylation and degradation of cactus, nuclear accumulation of Dorsal, involvement of MyD88, etc). Thus, although it is possible that Toll-mediated antiviral responses contribute to the phenotypes under investigation, considerably more data is needed to support that contention.

2. Statistical evaluations: The authors have gone to considerable lengths to stain, image, and quantify GSCs in dissected ovaries. This is a truly impressive accomplishment, and has likely generated some very interesting data. However, I am concerned about the statistical methods they have used to quantify their results. In particular, I am concerned about the tests used to evaluate data derived from counting GSCs per ovary. For this part of the study, the authors generated large amounts of categorical data where each ovary was scored as having 0, 1, 2, or 3-4 GSCs. It is not at all clear why 3-4 were grouped in a single bin. Surely, they should also be scored separately? More importantly, it is not clear why the authors chose t-tests for the comparisons between the groups, as the relevant section of the materials and methods is very short. For this reviewer, the t-test is inappropriate for almost all comparisons performed in this manuscript. The authors should be encouraged to re-consider their statistical approaches to all assays presented in the manuscript, relying on more appropriate tests, and writing a much more detailed methods section that allows readers to determine whether the tests are indeed appropriate.

3. Presentation of data: On a minor note, it is not advisable to present multiple data measurements as bar charts with error bars. The authors should be encouraged to use any one of numerous charting methods to properly lay out all data from all assays (box plots/dot plots/violin plots/etc). More importantly, many of the figures do not present data for relevant and important controls. To provide one example, figure 3 compares the effects of piwi-RNAi to a combination of piwi-RNAi and sgg-RNAi. Importantly, the piwi-RNAi flies should express a control RNAi construct (e.g. GFP-RNAi) to confirm that the rescue is not simply a result of diminishing RNAi-dependent depletion of the piwi transcript through the introduction of an extra construct. Similarly, many phenotypes are only reported in combination with the piwi-RNAi phenotypes with no data on the effects of single gene inactivation (e.g. Toll RNAi phenotypes).

4. Phenotypic effect and variability: Much of the quantitative data is interpreted based on p-values. However, for this reviewer the effect of piwi loss appears quite mild in many instances. The authors should be encouraged to include size effects when making claims of significance. Of equal importance, the authors should be asked to address the apparent variability of the piwi-RNAi phenotype in the manuscript. For example, Figure 1d shows an apparently highly significant loss of GSCs in piwi-RNAi flies by two weeks. However, in figure 2e, this effect appears to be gone for piwi-RNAi2. Specifically, a visual inspection of the first column (bab1>GFP-RNAi, no 3-TC), and the fifth column (bab1>piwi-RNAi2, no 3-TC) suggests that there are no differences between the two genotypes. Likewise, the 7th column (bab1>piwi-RNAi2) appears distinct from the 5th. Similar effects are apparent in supplementary figure 7. Specifically, the %GSC remaining data for 2261>GFP raised for 7 weeks in the absence of RU486 (panel d second column), are quite different to the percentage of GSCs remaining in 2261>armS10 raised for 7 weeks in the absence of RU486 (panel e, 2nd column). This variability does not necessarily weaken the claims of the

manuscript, and there are several possible explanations (genetic background etc). Nonetheless, it would be helpful for the authors to acknowledge this point in their manuscript, and to provide frequent descriptions of size effects, so that readers can appreciate the extent of variability.

5. Supplementary data: For this reviewer, the possible links to Alzheimer's models is not particularly relevant to the report, and seems distracting and premature. I would encourage the authors to consider removing these supplemental data so that they can focus on an interesting GSC phenotype.

<Response to reviewers>

We thank all of the reviewers for all of their critical and constructive comments.

Reviewer #1 (Remarks to the Author):

The manuscript of Lin et al. addresses Overall, results reported here could potentially interest a broad audience. However, there are several major concerns that need to be addressed.

Major concerns:

1. The link between toll signaling and retrotransposon de-repression requires further support. The authors showed that anti-RT drug reduces the number of viral-like particles. However, the link between viral-like particles and toll pathway is not based on experiments in the paper and instead relies on previous observations in mammals (citations 42-44). More work is needed to demonstrate causation relationship between expression of retrotransposons and toll signaling in Drosophila. For example, can over-expression of gypsy activate toll signaling?

<Response to reviewer>

We thank the reviewer for this comment. We believe that together, our genetic studies strongly support a causal relationship between a single pathway with *gypsy* and Toll and disruption of GSC maintenance. Overall, our data show that decreasing Piwi expression in cap cells activates retrotransposons and GSK3, which causes degradation of β -catenin, leading to a decrease of E-cadherin-mediated GSC attachment to cap cells. To the reviewer's specific point, either suppressing retrotransposon expression (by *gypsy* knockdown or inhibition of retrotransposon reverse transcription) or decreasing Toll expression in *piwi*-KD cap cells are sufficient to nearly completely rescue the phenotype (to wild-type levels). These results suggest that the observed retrotransposon derepression and Toll signaling activation are involved in the same pathway.

The most direct evidence to demonstrate a link between virus-like particles and Toll signaling would be to infect cap cells with isolated virus-like particles and examine Toll signaling. However, this experiment is extremely technically difficult, and we have not successfully performed it. As suggested by the reviewer, we could also overexpress retrotransposons (e.g., *gypsy*) in cap cells and examine Toll signaling. However, overexpression experiments are also technically difficult due to a lack of existing tools for overexpressing *gypsy*, and the fact that *gypsy* transcripts are targeted by Piwi for transcriptional silencing.

Since we are not able to provide molecular evidence for a direct link between retrotransposons and Toll signaling, we have softened our title from

“Transposon-activated Immune Signaling in the Aged Niche Eliminates Germline Stem Cells” to “Retrotransposon Derepression in the aged niche Eliminates Germline Stem Cells via Toll-mediated signaling”.

Also, it is not obvious how a drug that targets reverse transcriptase can reduce the number of viral particles.

<Response to reviewer>

Retrotransposon transcripts in the cytoplasm undergo reverse transcription to make retrotransposon cDNA, which then integrate into the genome for retrotransposon replication. Treatment with lamivudine (3TC; an analogue of cytidine) suppresses reverse transcription for retrotransposon replication, thus decreasing materials for making retrotransposon transcripts which encode viral protein for viral particle assembly.

*2. The claim that Piwi plays an important role during aging is compromised by the inability to rescue the phenotypes by Piwi supplementation (Supp.Fig. 9). It is unclear why authors resorted to a sophisticated drug-induced over-expression of Piwi. Can constitutively overexpressing piwi by *bab1>piwi* prevent/slow aging? It is critical to demonstrate that if Piwi is not lost in cap cells during the 5-week period, aging phenotypes can be circumvented.*

<Response to reviewer>

We thank the reviewer for this comment. As suggested, we continuously overexpressed Piwi in cap cells, finding that the GSC number is maintained better in Piwi-expressing flies than controls over time. We have added this new result in Supplementary fig. 11 and briefly described it on pg. 10, line 13. We also think the reviewer may have misinterpreted our original results showing that supplementation of Piwi in aged cap cells did indeed rescue the phenotype, significantly restoring Arm expression and E-cadherin distribution to the cap cell-GSC junction, as well as delaying age-dependent GSC loss. Notably, E-cadherin expression in the aged niche was not rescued by Piwi supplementation, probably due to the involvement of other factors (discussed on pg. 11, line 7). Moreover, we chose the inducible system to supply Piwi in the niche in order to avoid confounding effects caused by continuous overexpression of Piwi, and to demonstrate the requirement for Piwi is specific to aged cap cells. We have added a short description better explaining our rationale for this experiment in the main text, pg. 10, line 14.

3. It is unclear why a single retrotransposon, gypsy, plays a major role in aging. According to author results, specific knock-down of gypsy expression is sufficient to

rescue the loss of GSCs caused by Piwi depletion. This result indicates that other transposons do not play a significant role. However, other elements such as mdg1 are upregulated more strongly (210-fold) upon Piwi depletion in OSC cells compared to gypsy. Given that knock-down of gypsy can rescue the phenotype to the same degree as the application of an anti-RT drug that supposedly reduces the activity of all retrotransposons, what is special about gypsy that is different from all other transposons? How much do other transposons (retro- and DNA transposons) contribute?

<Response to reviewer>

We thank the reviewer for these questions. The *gypsy* retrotransposon is a highly active natural transposon in *Drosophila* (Song et al., 1994, *Genes and Development*; Krug et al., 2017, *PLOS Genetics*), which encodes an endogenous retrovirus with similar function to Human endogenous retrovirus K (expressed in some patients with amyotrophic lateral sclerosis) (Douville et al., 2011, *Annals of neurology*; Li et al., 2015, *Science Translational Medicine*). Moreover, its replication and generation of *de novo* insertions have been previously demonstrated in aged fly brain (Li et al., 2013, *Nature Neuroscience*). We believe this strong expression, function and aging-related modulation could explain why this single retrotransposon may be responsible for the GSC loss phenotype when *piwi* is depleted in cap cells. Coincidentally, neurodegeneration in a fly TPD-43 model is rescued by *gypsy* knockdown. We have added more description of the characteristics that make *gypsy* unique on pg. 7, line 7.

We also attempted to rescue the *piwi*-KD phenotype by knocking down another retrotransposon, *copA*; however, *copA* knockdown was unable to rescue the GSC-loss phenotype. As for the other retrotransposons increased in *piwi*-KD OSCs, we do not have suitable RNAi lines to test their roles. In our genomic sequence analysis, we did identify new insertion sites generated by DNA transposons and retrotransposons; however, we believe that testing whether these insertions disrupt gene function and the effects on GSC maintenance are beyond the scope of our current study. Overall, we show that *gypsy* upregulation is at least largely responsible for the GSC-loss phenotype, but our results do not completely rule out roles for other retrotransposons on the GSC-loss phenotype.

4. *The cytoplasmic role of Piwi remains unknown, so it is difficult to interpret results from Piwi [NT] mutant that localizes exclusively in the cytoplasm or to explain how it maintains GSC. It has not been shown that cytoplasmic Piwi targets transposon transcripts for degradation as authors claimed (line 95, 98). The endonuclease activity of Piwi is not required for any function in vivo (Darricarrère et al., PNAS 2013).*

Without elucidating the molecular function of cytoplasmic Piwi, authors cannot comment on how cytoplasmic Piwi maintains GSC or silence retrotransposons.

<Response to reviewer>

We thank the reviewer for these comments pointing out our potential mis/over-interpretation of data from previous studies. We have completely removed the *piwi^{NT}* data and conclusion regarding the cytoplasmic role of Piwi in retrotransposon silencing, as it is not our major focus (also see our response to reviewer 2, comment 3).

Of note, evidence from several studies led us to suggest that cytoplasmic Piwi may target transposon transcripts for degradation. First, Piwi contains a Piwi domain, which shows endonuclease activity on RNA *in vitro* (Saito et al., 2006, Genes and Development). Second, in the study done by Darricarrère et al. (PNAS 2013), Piwi with catalytic mutation still localizes to the nucleus, where it epigenetically suppresses TE transcription. As expected, transposon silencing and germ cell homeostasis are not affected. Therefore, this study cannot totally rule out a role for cytoplasmic Piwi in TE or mRNA transcript degradation. We suspect that nuclear Piwi may act as a first line of defense to suppress TEs. Third, upregulation of somatic transposon expression (*gypsy* and *mdg1*) in *piwi^{NT}* mutants is only about half of that in *piwi²* null mutants (See Fig. 3A, Klenov et al, 2011, PNAS). We would expect to see the same degree of increase in transposon expression if Piwi completely blocks transposon transcription. Thus, *piwi^{NT}* may not completely block its nuclear entry, or cytoplasmic Piwi may serve as a second defense line for TE activation. As *Piwi^{NT}* is nearly undetectable in the nucleus, we prefer the later possibility.

Minor concerns:

1. *Authors should describe the usage of Gal80ts and temperature shift in the main text to explain why KD starts after eclosion (line 57) and why studying bab1>piwiRNAi in 2W.*

<Response to reviewer>

We thank the reviewer for this suggestion. We previously described the usage of GAL80ts in the Materials and Methods, and we now have added a short description in the main text on pg. 3, line 17. Knockdown of *piwi* in CpCs for one week did not decrease Piwi expression, probably due to persistence of the protein. On the other hand, Piwi was nearly absent in the CpCs of 2-week-old *piwi*-KD flies. Therefore, most of the phenotypes for *piwi*-KD CpCs were characterized at 2 weeks of age. We have added a short description on pg. 4, line 1.

2. *Phenotypes of all double KD will be more convincing if the control is co-KD an*

unrelated gene like gfp, instead of KD piwi alone where 2x as much drives goes to piwiRNAi.

<Response to reviewer>

We thank the reviewer for this comment. For the rescue experiments using double knockdowns (Fig. 3h, j, g and Fig. 4a), we performed co-knockdown of *piwi* and *gfp* simultaneously as the control. In this revised manuscript, we have added a clear genotype description to each panel.

However, we only used *piwi*^{RNAi} alone as the control for the *piwi*^{RNAi-2} & *gypsy*^{RNAi} co-knockdown experiments (Fig. 2e, Fig. 3h and Fig. 4b). The GSC numbers in cap cells with *piwi*^{RNAi-2} & *copia*^{RNAi} (Fig. 2e) and *piwi*^{RNAi-2} & *loki*^{RNAi} co-knockdown flies (Supplementary fig. 5c) were comparable to *piwi* knockdown alone, suggesting that rescue of the GSC loss phenotype in the germlaria carrying *piwi*-KD cap cells by suppressing *gypsy* expression is not due to the splitting of GAL4 activity by two UAS elements.

3. It helps if authors can describe levels of F-actin, toll signaling and pSer9-GSK9 in aged cap cells, in addition to the KD model, to argue that they occur during aging.

<Response to reviewer>

We thank the reviewer for these suggestions. In this revised manuscript, we added new results showing decreases of F-actin (Supplementary Fig. 8 e and e') and pSer9-GSK3 levels (Supplementary Fig. 10a and a') in aged cap cells. We also briefly described the results on pg. 9, line 5 and supplementary Fig. 10)

Upon Toll signaling activation, Cactus (orthologue of IκB in mammals) is phosphorylated and degraded in the cytoplasm to release Dorsal (orthologue of NF-κB in mammals); thereafter, Dorsal enters into the nucleus to activate transcription of genes encoding Antimicrobial peptides (AMPs). In our results, Cactus and Dorsal were expressed in the cytoplasm of *bab1>gfp*^{RNAi} cap cells, while Cactus and Dorsal were both absent from the cytoplasm of cap cells with overexpression of constitutively active Toll^{10b}. All known AMPs were expressed at undetectable levels in our cap cell transcriptome analysis. Perhaps this result was due to amplification bias in the cDNA made from isolated cap cells, our examination of unregulated AMPs, or quick removal of Dorsal from the nucleus. We therefore used the absence of Cactus in the cytoplasm as an indicator of Toll signaling activation in cap cells. Our results show that Cactus is also dramatically reduced in aged cap cells; however, we cannot claim this reduction is due to activation of Toll signaling.

4. It would make the manuscript much stronger if authors do RNA-seq in sorted niche cells. OSC can serve as a reference, but it describes other cell types like follicular cells.

<Response to reviewer>

We understand the reviewer's concern. While OSCs express follicle cell markers, these cells also exhibit niche characteristics by expressing Dpp, which can support cultured ovarian germline stem cells (Niki et al., PNAS, 2006). Therefore, the transcriptome analysis in *piwi*-KD OSCs can provide useful information for our study. However, we also noticed that the transcriptome analysis in OSCs was derived from only one replicate; therefore, we have completely removed the OSC data from the current version of the manuscript.

As suggested by the reviewer, we isolated niche cells from *bab1>gfp & piwi^{RNAi-1}* germaria for RNA sequencing of biological replicates [Note that our isolated niche cells not only have cap cells, but also include terminal filament cells, as they also express GFP but with extremely weak Piwi expression (Ma et al., 2014, PLOS one; Szakmary et al., 2015, Current Biology)]. Our results show the following:

(i) Piwi expression in the RNAi line was reduced to 40% of the level in isolated control cap cells (*bab1>gfp*). Consistent with the transcriptome data from *piwi*-KD OSCs, expression levels for a number of retrotransposons were increased (including *gypsy* and *mdg1*), although the expression fold change for each retrotransposon was lower than that in the OSC transcriptome analysis. We have added this result as Fig. 2c, and described the results in the main text, pg. 5, line 12.

(ii) Unlike the transcriptome data obtained from *piwi*-KD OSCs, almost all the known antimicrobial peptides (AMPs; Toll downstream targets, such as Drosomycin) were undetectable in isolated niche cells. We suspect that this inability to detect AMP expression may be due to the bias from cDNA amplification using a limited number of isolated cap cells [only 4-6 cap cells per germarium], quickly removal of Dorsal, or these AMPs might not be expressed in CpCs.

5. *Can there be aging-dependent, piwi-independent changes in E-cad/Arm/F-actin, GSK3 phosphorylation state, toll signaling? Complexity of aging could be discussed more.*

<Response to reviewers>

We thank the reviewer for this question and comment. Yes, there are aging-dependent, *Piwi*-independent changes in E-cad/Arm/F-actin expression. Interestingly, supplementation of Piwi in the aged niche could not restore total E-cad expression levels, indicating that Piwi does not control E-cad expression levels in the aged GSCs. Moreover, Notch signaling was reported to be elevated in aged GSCs, which suppresses E-cad expression in the GSC-cap cell junction (Tseng et al., PLOS Genetics, 2014). When E-cad expression is reduced, expression/or localization of Arm

and F-actin may also be affected, as the intracellular domain of E-cadherin binds Arm, and Arm binds F-actin.

Since GSK3 is regulated by phosphorylation, which is carried out by different kinases, and Toll signaling is activated by bacteria and fungi, we believe GSK3 phosphorylation state and Toll signaling are also regulated by Piwi-independent mechanisms. We have added a brief discussion regarding the complexity of these aging-related phenotypes in the main text, pg. 11, line 7 and supplementary Fig. 10.

6. Can 3TC prevent RT in GSCs as well to at least in part explain the rescue?

<Response to reviewers>

We thank the reviewer for this question. It is possible, but less likely, that 3TC prevents reverse transcription (RT) in GSCs, unless viral materials encoded by retrotransposons are transported/or present in GSCs. However, we did not find virus-like particles in GSCs by TEM (see Fig. 5).

7. AD data in the end seem largely out of place. Adding these weakens the manuscript.

<Response to reviewers>

We understand the reviewer's concern, which is why we present these results as supplementary information. We would like to use these data to demonstrate the relevance of retrotransposon-GSK actions to other aging-related conditions. Although the results are not highly related to GSCs, we believe this finding "retrotransposon-GSK3 regulation" could benefit research on aging-associated tissue degeneration (e.g., AD). In this revised manuscript, we therefore still kept these findings as supplementary information.

8. In Fig S9e, 7W 2261 without induction does not have aging-related GSC loss (~90% GSCs left). GSC loss during aging does not appear robust when compared with Fig 1d.

<Response to reviewers>

We thank the reviewer for this question. The GeneSwitch system is known to have leaky GAL4 activity in the absence of RU486 (Ke and Hsu, 2019, G3; Scialo et al., 2016, PLOS one). The less robust GSC loss during aging in 2261>Arm^{S10} and HA-Piwi in the absence of RU486 induction may be caused by leaky expression of Arm^{S10} and Piwi; nevertheless, the GSC number is significantly increased when adding RU486 for one week. We have added a brief note about leaky regulation in these lines to the main text, pg. 10 line 19, and in the legend of Supplementary fig. 12d.

9. "niche" and "cap cells" are used somewhat interchangeably throughout the text.

Authors should be more accurate when describing effects seen in cap cells only.

<Response to reviewers>

We thank the reviewer for this suggestion. We have modified the text to specifically mention cap cells when the analysis was performed in cap cells.

10. Line 178, will EC (in addition to TF+CpC) also be GFP-positive?

<Response to reviewers>

We thank the reviewer for this question. Only the anterior-most ECs, which contact GSCs, express very weak GFP (Supplementary fig. 3a).

11. Method of retrotransposon analysis is poorly documented. For instance, which annotation is used? e.g. springer cannot be found in either RepBase of FlyBase.

<Response to reviewers>

We thank the reviewer for this comment. We have added more information to our description of retrotransposon analysis in the Materials and Methods, pg. 29, line 5. We used the “Transposable elements (canonical set)” database downloaded from FlyBase for transposon annotation; the ID of Springer in FlyBase is FBte0000333.

12. Fig 2c, pie chart is an ineffective way to visualize data, where one cannot appreciate a de-repression of transposon without reading the numbers.

<Response to reviewers>

We thank the reviewer for this comment. We have changed the pie chart to a scatter plot to better show TE expression levels in *piwi*-KD niche cells (Fig. 2c).

13. Nearly all “enlarged” images lack scales. Also, in Fig S7b, outline in enlarged image seems highly inconsistent with the outline in the original image.

<Response to reviewers>

We thank the reviewer for this comment. We have added scale bars in the enlarged images and mentioned these scale bar sizes in the legends of figures 2, 3 and 4. We also modified Fig. S12b (original S7b) to make the enlarged image and original image consistent.

14. In Fig 3b bottom row, it is not obvious that there are more than one LamC+ CpC. A more representative image should be shown.

<Response to reviewers>

We thank the reviewer for this comment. We have replaced the original image with a more representative one.

15. While cap cells survive by 2W, there is statistically significant increase of DNA damage (Fig S4), which should be described in text.

<Response to reviewers>

We thank the reviewer for this comment. We have added a brief statement that DNA damage is significantly increased in 2- and 5-week-old cap cells, in the main text, pg. 7, line 16 and Supplementary fig. 4b'

16. Image segmentation and quantification are not sufficiently documented. e.g. are cells outlined manually? Is it reliable to use nuclear marks (tj, piwi, lamC) to outline cell boundary? For analysis at cell-cell boundary that is 3D, is 2D quantification reliable?

<Response to reviewers>

We thank the reviewer for this comment. We have added a more thorough description of how we did the quantification in the Materials and Methods, pg. 22, line 17. We did outline cells manually. Cap cells have very thin cytoplasm, and thus their plasma membranes and nuclei are very close to each other [an example is shown in Fig. 3a, E-cad expressed on the plasma membrane is adjacent to Lam C signal, which labels the nuclear envelope]. Thus, we believe that using a nuclear marker to outline cap cells is reliable, although it is not an ideal solution. For analysis at the cell-cell boundary, we measured the intensity but not the area, choosing the section with the highest expression of the protein of interest for measurement.

Reviewer #2 (Remarks to the Author):

In Drosophila females, the identity of the adult germ line stem cells (GSCs) is maintained by the In summary, the results presented in the manuscript may be of interest to a wide range of readers

Critical comments:

1) *The authors observed, and quoted the literature, that number of GCS decreased with age. They also showed that piwiRNAi in CpCs was resulting in a decrease in GCS number. However, two weeks were needed for the complete loss of Piwi expression (line 57). The two weeks lag of the effect of RNAi could well be the result of the well-known protein perdurance. In line 61, the authors wrote: "As previously reported^{12,13}, the number of GSCs in control flies decreased with age (fig. 1d and Supplementary Table 1). This decrease was accelerated in two independent piwi-knockdown (KD) lines (fig. 1d and 64 Supplementary Table 1)." The authors then consistently, but I think wrongly, classify piwi silenced phenotype as an age dependent loss." In my opinion, if RNAi had an immediate effect on Piwi protein depletion, the authors would not write about an increase of age-dependent loss. It*

would be better to apply the term age-dependent loss only to the phenotypes measured in the aged wild type flies.

<Response to reviewers>

We understand the reviewer's comment. However, we measured GSCs in flies at ages: day 1, 2 weeks, and 5 weeks. GSC numbers in control flies from 2 weeks to 5 weeks of age dropped from 83% to 69% of day one levels (a 14% reduction). Regardless of whether GSCs were maintained by Piwi perdurance before 2 weeks of age, flies with depleted Piwi expression in the niche showed a 21-23% reduction of GSCs from 2 to 5 weeks using either of two different *piwi* RNAi lines. These results clearly show that *piwi* depletion in cap cells accelerates the age-dependent GSC loss. We, therefore, think that our conclusion is accurate and have kept our description.

2) *The authors clearly demonstrated that the piwi mediated protection from the harmful effect of Gypsy retrotransposon is strictly cap cell specific. The other two types of somatic niche cells, the terminal filament cells and the anterior escort cells are not involved in this process. TF cells do not express Piwi at all. Piwi expression is only slightly reduced in the very old, 8-week-old escort cells. To examine which types of retrotransposons are suppressed by Piwi in the niche cells, the authors examined a publicly available transcriptome database for retrotransposon expression levels in a piwi-knock down ovarian somatic cell line (OSC). In line 85 the authors wrote ".....ovarian somatic cells (OSCs), which resemble CpCs to support GSCs 21." According to the citation no 21: „We assume that the somatic component of the cultures (from which the OSC cell lines were sub cloned) is derived from the somatic stem cells (cf. Fig. 1b)". This means that the origin of the OSC cell line is ambiguous and the used cell lines most probably do not represent CpCs properly. Therefore, I suggest to omit the in silico analysis of OSC expression patterns from the manuscript.*

<Response to reviewers>

We agree with the reviewer's comment. In this revised manuscript, we have removed the OSC transcriptome analysis data. As suggested by the reviewer 1, we have isolated biological replicates of *piwi-KD* and *gfp-KD* niche cells (containing both TF cells and CpCs) for RNA sequencing. Notably, our isolated niche cells also include terminal filament cells, as they also express GFP but with extremely weak Piwi expression (Ma et al., 2014, PLOS one; Szakmary et al., 2015, Current Biology). Our results show:

(i) Piwi expression in the RNAi line was reduced to 40% of the level in isolated control cap cells (*bab1>gfp*). Consistent with the transcriptome data from *piwi-KD* OSCs, expression levels for a number of retrotransposons were increased (including *gypsy*

and *mdg1*), although the expression fold change for each retrotransposon was lower than that in the OSC transcriptome analysis. We have added this result as Fig. 2c, and described the results in the main text, pg. 5, line 12.

(ii) Unlike the transcriptome data obtained from *piwi*-KD OSCs, almost all the known antimicrobial peptides (AMPs; Toll downstream targets, such as Drosomycin) were undetectable in isolated niche cells. We suspect that this inability to detect AMP expression may be due to the bias from cDNA amplification using a limited number of isolated cap cells (only 4-6 cap cells per gerarium), quickly removal of Dorsal from the nucleus, or these AMPs might not be expressed in CpCs. In addition, we noticed that the transcriptome analysis results from OSCs were from only one replicate. Therefore we completely removed OSC data from the current revised manuscript.

3) line 94 *“Piwi acts in the nucleus through epigenetic mechanisms to suppress transposon transcription²², while cytoplasmic Piwi targets transposon transcripts for degradation²³. Ovaries carrying mutant Piwi with a nuclear localization defect (*piwi^{NT}*) did not accelerate age-dependent GSC loss (Supplementary Fig. 3 and Supplementary Table1), suggesting that Piwi suppresses cytoplasmic retrotransposon transcripts for replication acting as the second defence mechanism to maintain GSCs.”*

*Firstly, it seems to me that ref. 23 was wrongly cited. It is a general though in the field, that Piwi exclusively participates in co-transcriptional silencing that happens in the nucleus and does not participate in the cytoplasmic silencing. It is also generally accepted that in the ovarian somatic cells only the co-transcription silencing but not the cytoplasmic silencing is functional. Secondly, The cited ref. 23 says: “Transcript level of the Gypsy and *mdg1* retrotransposons increased 10- to 20-fold in the ovaries of homozygous *piwi^{NT}* and *piwi2* or transheterozygous *piwi2/piwi^{NT}* flies relative to the corresponding heterozygotes (Fig. 3A).” Later: “Here we describe the phenotype of a unique mutation in the *piwi* gene that leads to the formation of cytoplasmic Piwi^{NT} (i.e., N-truncated Piwi protein) lacking the NLS. The properties of this mutant made the direct influence of the piRNA pathway on GSC maintenance unlikely, as the *piwi^{NT}* mutant displayed normal GSC self-renewal (Fig. 1 C and D) but lost Piwi mediated transposon repression completely in ovarian cells (Fig.3A) including niche cells responsible for GSC self-renewal signalling (Fig. 4B). “*

<Response to reviewer>

We thank the reviewer for these comments and understand their meaning. We have completely removed the *piwi^{NT}* data and any mention of the cytoplasmic role of Piwi

in retrotransposon silencing, as it is not our major focus.

Evidence from several studies led us to suggest that cytoplasmic Piwi may target transposon transcripts for degradation. First, Piwi contains a Piwi domain, which shows endonuclease activity on RNA *in vitro* (Saito et al., 2006, Genes and Development). Second, in the study done by Darricarrère et al. (PNAS 2013), Piwi with catalytic mutation still localizes to the nucleus, where it epigenetically suppresses TE transcription. As expected, transposon silencing and germ cell homeostasis are not affected. Therefore, this study cannot totally rule out a role for cytoplasmic Piwi in TE or mRNA transcript degradation. We suspect that nuclear Piwi may act as a first line of defense to suppress TEs. Third, upregulation of somatic transposon expression (*gypsy* and *mdg1*) in *piwi^{NT}* mutants is only about half of that in *piwi^{1/2}* null mutants (See Fig. 3A, Klenov et al, 2011, PNAS). We would expect to see the same degree of increase in transposon expression if Piwi completely blocks transposon transcription. Thus, *piwi^{NT}* may not completely block its nuclear entry, or cytoplasmic Piwi may serve as a second defense line for TE activation. As *piwi^{NT}* is nearly undetectable in the nucleus, we prefer the later possibility.

There is a plain contradiction between the ref. 23 and the manuscript. According ref. 23, there is an elevated Gypsy level in piwi^{Nt} mutant germlaria (most probably in young ovaries since the ref. 23 did not focus on the ageing). According the manuscript, however, "Ovaries carrying mutant Piwi with a nuclear localization defect (piwi^{NT}) did not accelerate age-dependent GSC loss." With the other words the elevated Gypsy level in the GSC niche might not be the reason of the age dependent GSC loss. This contradiction must be resolved in every way.

<Response to reviewer>

We thank the reviewer for raising this important issue. We consider two possible explanations for this contradiction:

(i) Production of *gypsy* may occur in follicle cells of *piwi^{NT}* mutants. Piwi not only suppresses retrotransposons in cap cells, but also in escort cells and follicle cells (Malone et al., 2009, Cell; Ma et al., 2014, PLOS one). Notably, the ovaries of *piwi²* mutants are deformed and very small, with hardly any egg chambers (see Fig. 1B in Klein et al., 2016, PLOS Genetics). Conversely, *piwi^{NT}* mutant ovaries carry egg chambers (one ovariole carries 4.6 egg chambers, and egg chambers after stage 7 contain ~650 follicle cells) (Klenov et al., 2011, PNAS; and our observation). A wild-type ovary is composed of 16-20 ovarioles, with each germarium carrying 5-7 cap cells and about 30 escort cells. Presumably a *piwi²* mutant ovary carries 80 cap cells, 480 escort cells, and 20,800 follicle cells (we assume only 2 egg chambers older

than stage 7); therefore, by this estimation, the follicle cell number is 37-fold higher than the number of cap cells plus escort cells. Thus, the major *gypsy*-expressing cell types in *piwi*² and *piwi*^{NT} mutant ovaries may well be different.

(ii) Cytoplasmic Piwi might function to degrade retrotransposons. If TE transcription is completely activated in *piwi*^{NT} mutants, we would expect to see higher expression of *gypsy* in *piwi*^{NT} ovaries as compared to *piwi*² mutant ovaries, as follicle cells also express high level of *gypsy* (see description above). However, *gypsy* expression in *piwi*^{NT} is only half of that in *piwi*² (See Fig. 3A, in Klenov et al., 2009, PNAS), which is consistent with the idea that suppression of retrotransposon transcription does not occur in the *piwi*^{NT} mutants. Thus, either Piwi^{NT} does not completely block its nuclear entry, or cytoplasmic Piwi serves as a second line of defense against TE activation. As Piwi^{NT} is nearly absent from the nucleus, we prefer the later possibility. We now have added a short discussion of this contradicting result in the main text, pg. 14, line 17.

In the revised manuscript, we also generated cap cells with *piwi* homozygous mutation by genetic recombination, and we found the homozygous mutant cap cells display dramatically reduced Armadillo (β -catenin orthologue in the fly) at the junction with adjacent GSCs. We have added this result in supplementary fig. 6 and briefly describe it on pg. 8, line 18. This result, together with the results of our *piwi*-knockdown, *piwi*-knockdown plus 3TC treatment, and *piwi* & *gypsy* co-knockdown experiments strongly support the idea that retrotransposon reactivation causes GSC loss during aging. Furthermore, we have completely removed *piwi*^{NT} mutant data from the manuscript, as the function of Piwi in the cytoplasm is not our focus.

4) line 98: “.....suggesting that Piwi suppresses cytoplasmic retrotransposon transcripts for replication acting as the second defence mechanism to maintain GSCs. To test this hypothesis, we suppressed reverse transcription of replicating retrotransposons by feeding bab1>gfpRNAi-1 and bab1>piwiRNAi flies with (-)-L-2',3'-dideoxy-3'-thiacytidine (3TC), a cytidine analogue that is clinically used to inhibit reverse transcription of human immunodeficiency virus and Hepatitis B virus24,25.

If the cytoplasmic function of Piwi in transposon silencing had be true, the logic of the introduction of 3TC experiments would have been still incorrect. Namely, irrespective of whether the tertotransposon de-repression occurs in the nucleus or in the cytoplasm, 3TC experiment may give positive result. That is, by 3TC experiments it is not possible to test the hypothesis about the cytoplasmic activity of Piwi.

<Response to reviewer>

We agree with the reviewer's comment. We have removed our hypothesis regarding

the cytoplasmic role of Piwi in targeting transposon transcripts for degradation. Instead, we ask about the involvement of retrotransposons in the GSC loss phenotype by treating *piwi*-KD flies with 3TC; pg. 6, line 6.

5) line 40: "Despite their genomic abundance, transposons are effectively silenced by Piwi-interacting RNAs (piRNAs) at transcriptional and post-transcriptional levels 6,7. However, this process is known to be attenuated in aged tissues 8-10, where stem cells are frequently lost¹¹". In this context, "this process" refers to the piwi pathway. The authors refer three publications (No 8, 9, and 10) concerning the age-related attenuation of the Piwi system. Unfortunately, neither of the cited publications is about the Piwi pathway.

<Response to reviewer>

We agree with the reviewer's comments. We have modified the original sentence from "However, this process is known to be attenuated in aged tissues..." to "However, transposon silencing is known to be attenuated in aged tissue..."; pg. 3, line 5.

Reviewer #3 (Remarks to the Author):

This manuscript examines the age-dependent loss of germline stem cells in the Drosophila ovary. I have several concerns about claims contained in the manuscript. I have detailed my major concerns below.

1. Virus – Toll: In several instances, the manuscript discusses the generation of virus within the stem cell niche. These claims are not supported by the data. Unless the authors show that infectious cell-free virus particles capable of generating an integrated provirus in newly infected cells are produced, they cannot make any statements about virus production. The instances of virus-like particles in some TEMs are certainly interesting, but more work is required. Equally importantly, claims of virus-dependent activation of Toll signaling require more support.

<Response to reviewer>

We understand and agree with the reviewer's comment. Indeed, we are not able to demonstrate that the virus-like particles observed in TEM are truly virus, and we are not able to definitively claim that Toll signaling is activated by virus or to explain the molecular mechanism of how Toll receptors might detect the presumptive intracellular virus.

However, 3TC treatment, which suppresses reverse transcription, rescues all the phenotypes observed in the ovaries carrying *piwi*-KD cap cells, including expression of Arm, E-cadherin and pSer9GSK3, as well as GSC number. In addition, suppressing *gypsy* expression in the *piwi*-KD cap cells also rescued all the phenotypes.

Furthermore, simultaneously knocking down *piwi* and *toll* or *toll-5* also rescued GSC number and suppressed GSK3 activity. From these results, we can conclude that Piwi silences the *gypsy* retrotransposon in cap cells to maintain GSCs, and this effect requires Toll.

In this revised manuscript, we have softened our title from “Transposon-activated Immune Signaling in the Aged Niche Eliminates Germline Stem Cells” to “Retrotransposon Derepression in the Aged Niche Eliminates Germline Stem Cells via Toll-mediated Signaling”. In addition, we have also softened our description in the Abstract and main text; pg.14, line 6.

To support their claims that the niche is activating a Toll-dependent antiviral response, the authors cite two publications that link Toll signaling to antiviral defenses in Drosophila. However, it is important to note that one of those studies established a role for Toll-7 in antiviral defenses. Importantly, the authors effectively exclude Toll-7 as a receptor. The second cited manuscript implicated Toll in fly antiviral responses. However, the Toll receptor is generally considered to recognize microbial patterns at the plasma membrane.

In the context of this manuscript, it is not clear how Toll can detect cytosolic viral particles or genomes, nor do the authors present any data to indicate that is happening. It is possible that the putative antiviral responses under investigation here are being mediated by Toll-5, but there are no data to support an antiviral role for Toll-5, or to suggest that Toll-5 is able to bind viral particles or genomes in the cytosol.

<Response to reviewer>

We thank the reviewer for this critical comment. In mammals, TLR7 and TLR9 are located in the endosome and directly detect foreign DNA/RNA and proteins in the cytoplasm (Mogensen, 2009, Clin Microbiol Rev). Interestingly, *Drosophila* Toll is the homologue of TLR7/9 (supplementary table 4), as predicted by DIPOT (Hu, et al., 2011, BMC Bioinformatics). Toll is present in the plasma membrane and cytoplasm, and it requires the endocytic pathway for activation (Huang, et al., 2010, PNAS). Thus, Toll might behave similar to TLR7/9 in detecting viral genetic materials to activate immune signaling. On the other hand, it has been reported that Toll-5 interacts with Toll and participates in activation of Toll signaling (Luo, et al., 2001, Insect Mol Biol.). Toll-5 is the homologue of TLR4, which detects viral proteins (Mogensen, 2009, Clin Microbiol Rev); however, it is not clear if Toll-5 also detects viral proteins and its role in activating Toll signaling in cap cells is not confirmed. We have added a brief discussion regarding the possible mechanism for Toll/Toll-5 detection of viral materials on pg. 15, line 10.

Although we do not conclusively know that Toll/Toll-5 detect viral genetic material (which is beyond our scope at the moment), our data nevertheless show that Toll or Toll-5 knockdown in *piwi*-KD cap cells are able to rescue GSC number to the wild-type levels.

Furthermore, there are no data to support activation of the Toll pathway (e.g. phosphorylation and degradation of Cactus, nuclear accumulation of Dorsal, involvement of MyD88, etc).

<Response to reviewer>

We thank the reviewer for this comment. As suggested, we have performed IHC for Cactus and Dorsal in ovaries carrying *piwi*-KD cap cells. Cactus and Dorsal were expressed in the cytoplasm of *bab1>gfp^{RNAi}* cap cells; however, both Cactus and Dorsal were absent from the cytoplasm of cap cells from *piwi*-KD flies and when a constitutively active form of Toll, Toll^{10b}, was overexpressed. All known antimicrobial peptides (AMPs) were expressed at levels below the limit of detection according to our cap cells transcriptome analysis. This result may have been due to amplification bias of cDNA made from isolated cap cells, our probing of unregulated AMPs, or quick removal of Dorsal from the nucleus. Combined with the genetic data (decreased expression of Toll or Toll-5 in *piwi*-KD rescues GSC number and suppresses GSK3 activity) these data clearly show that Toll is required for retrotransposons to impair GSC maintenance.

Upon activation of the Toll signaling pathway, MyD88 serves as an adaptor to link Toll and downstream kinase Pelle, which phosphorylates Cactus for degradation, allowing Dorsal nuclear translocation to activate transcription of target genes that encode AMPs (Valanne, 2011, J Immunol). Drosomycin-GFP is not expressed in cap cells, and expression levels of AMPs in the transcriptome analysis of isolated *piwi*-KD cap cells were below detectable levels. In addition, decreased pSer9GSK3 in *piwi*-KD via Toll may be independent of MyD88. We, therefore, did not test if MyD88 is involved in Toll signaling.

Thus, although is possible that Toll-mediated antiviral responses contribute to the phenotypes under investigation, considerably more data is needed to support that contention.

<Response to reviewer>

We thank the reviewer for this evaluation. In the revised manuscript, we have softened the description regarding the link between Toll and virus; however, we believe that our data can fully support a model where retrotransposon derepression

in the aged niche requires Toll to reduce GSCs.

2. Statistical evaluations: The authors have gone to considerable lengths to stain, image, and quantify GSCs in dissected ovaries. This is a truly impressive accomplishment, and has likely generated some very interesting data. However, I am concerned about the statistical methods they have used to quantify their results. In particular, I am concerned about the tests used to evaluate data derived from counting GSCs per ovary. For this part of the study, the authors generated large amounts of categorical data where each ovary was scored as having 0, 1, 2, or 3-4 GSCs. It is not at all clear why 3-4 were grouped in a single bin. Surely, they should also be scored separately?

<Response to reviewer>

We thank the reviewer for this question. The number of germaria carrying 4 GSCs is very low in most groups (less than 10% of the total). Separating the groups of germaria carrying 4 GSCs from germaria carrying 3 GSCs does not change our conclusions. Thus, to simplify the graph we grouped germaria carrying 3 and 4 GSCs in the same category.

More importantly, it is not clear why the authors chose t-tests for the comparisons between the groups, as the relevant section of the materials and methods is very short. For this reviewer, the t-test is inappropriate for almost all comparisons performed in this manuscript. The authors should be encouraged to re-consider their statistical approaches to all assays presented in the manuscript, relying on more appropriate tests, and writing a much more detailed methods section that allows readers to determine whether the tests are indeed appropriate.

<Response to reviewer>

We thank the reviewer for this comment. In comparisons of two groups, we chose the t-test. We also performed chi-square tests to compare frequencies of events between two genotypes, but we omitted the results as they are similar to the t-test analyses. Nevertheless, as suggested by the reviewer, we have changed most of the statistical comparisons to One-way ANOVA (Fig. 3g, 3h, 3i, 3k and Fig. 4b', supplementary fig. 1b, c, d and i, supplementary fig. 8a', d' and e', supplementary fig. 10a', supplementary fig. 12f' and g', supplementary fig. 15b and d.). The comparisons were performed among multiple groups and are described in the statistical methods subsection of the Materials and Methods (pg. 24, line 6) and in the figure legends.

3. Presentation of data: On a minor note, it is not advisable to present multiple data measurements as bar charts with error bars. The authors should be encouraged to

use any one of numerous charting methods to properly lay out all data from all assays (box plots/dot plots/violin plots/etc). More importantly, many of the figures do not present data for relevant and important controls. To provide one example, figure 3 compares the effects of piwi-RNAi to a combination of piwi-RNAi and sgg-RNAi. Importantly, the piwi-RNAi flies should express a control RNAi construct (e.g. GFP-RNAi) to confirm that the rescue is not simply a result of diminishing RNAi-dependent depletion of the piwi transcript through the introduction of an extra construct. Similarly, many phenotypes are only reported in combination with the piwi-RNAi phenotypes with no data on the effects of single gene inactivation (e.g. Toll RNAi phenotypes).

<Response to reviewer>

We thank the reviewer for this comment. We have changed bar charts to Box plots or dot plots in Fig. 3 and 4 and supplementary fig. 1, 7, 8, 9, 10, 12 and 15. We still kept the original graphs showing GSCs and cap cell numbers, as it clearly shows the difference among germaria in response to the genetic manipulation.

We have also added proper controls for each experiment. For the rescue experiments using double knockdown (Fig. 3h, j, g and Fig. 4a), we performed co-knockdown of *piwi* and *gfp* simultaneously as the control. In this revised manuscript, we have added the genotype to each panel.

We only used *piwi*^{RNAi} alone as the control for *piwi*^{RNAi-2} & *gypsy*^{RNAi} co-knockdown experiments (Fig. 2e, Fig. 3h and Fig. 4b) because the GSC number in the cap cells of *piwi*^{RNAi-2} & *copid*^{RNAi} (Fig. 2e) and *piwi*^{RNAi-2} & *loki*^{RNAi} co-knockdown flies (Supplementary fig. 5c) were comparable to *piwi* knockdown alone. These results indicate that rescue of GSC loss phenotype in the germaria carrying *piwi*-KD cap cells by suppressing *gypsy* expression is not due to splitting of GAL4 activity between two UAS elements.

We also added results of GSC analysis in single gene knockdown flies, including *sgg*-, *toll*-, *toll-5*, and *toll-7*-KD (Supplementary Table 1). We found that depletion of Toll-5 in the niche for 2 weeks reduced GSC number by 17% compared to *gfp*-KD control, while depletion of *sgg*, *toll* and *toll-7* did not affect GSC number at 2 weeks. We do not know why knockdown *toll-5* alone caused a reduction of GSCs.

3. Phenotypic effect and variability: Much of the quantitative data is interpreted based on p-values. However, for this reviewer the effect of piwi loss appears quite mild in many instances. The authors should be encouraged to include size effects when making claims of significance.

<Response to reviewer>

We thank the reviewer for this comment. In the control group, the normal rate of

GSC loss from flies at Day 1 to 2 weeks old was approximately 17-20%. This loss rate was increased to 25-34% in flies carrying *piwi*-KD cap cells (either using *piwi*^{RNAi-1} or *piwi*^{RNAi-2}). This increased loss was similar to the rate observed in the control at 5 weeks of age. Although the GSC loss rate is maximally increased only about 2-fold compared to controls (34% vs. 17%), deletion of Piwi in cap cells speeds up the aging phenotype of GSC loss by 3 weeks. We now have added a short description on pg. 4, line 12 to better explain the significance of Piwi in cap cells for GSC maintenance.

To increase the sample size for each experiment, we randomly collected 20 flies of each genotype for dissection. We then examined a few germaria from each ovary; the total number of examined germaria for each genotype was about 100. We now have added this statement in the Materials and Methods section, pg. 23, line 19.

Of equal importance, the authors should be asked to address the apparent variability of the piwi-RNAi phenotype in the manuscript. For example, Figure 1d shows an apparently highly significant loss of GSCs in piwi-RNAi flies by two weeks. However, in figure 2e, this effect appears to be gone for piwi-RNAi2. Specifically, a visual inspection of the first column (bab1>GFP-RNAi, no 3-TC), and the fifth column (bab1>piwi-RNAi2, no 3-TC) suggests that there are no differences between the two genotypes. Likewise, the 7th column (bab1>piwi-RNAi2) appears distinct from the 5th.

<Response to reviewer>

We thank the reviewer for this comment. *piwi*^{RNAi-2} did give us a milder phenotype, in terms of GSC loss. We now have mentioned this point in the main text, pg. 6, line 13. The GSC loss rate in *bab1>piwi*^{RNAi-2} flies (-3TC) from Day 1 to 2 weeks was about 26% (5th column) and 32% (7th column), which falls into the range of 25-34% described above. In fig. 2e, in fact GSC maintenance in *gfp*-KD controls without 3TC (1st column) treatment is significantly higher than *bab1>piwi*^{RNAi-2} (-3TC, the 5th column). To avoid confusion with visual inspection of our graph, we now have added statistical analysis results between genotypes on the graph, e.g., *gfp*-KD (-3TC) vs. *piwi*-KD (-3TC) in Fig. 2e.

Similar effects are apparent in supplementary figure 7. Specifically, the %GSC remaining data for 2261>GFP raised for 7 weeks in the absence of RU486 (panel d second column), are quite different to the percentage of GSCs remaining in 2261>armS10 raised for 7 weeks in the absence of RU486 (panel e, 2nd column). This variability does not necessarily weaken the claims of the manuscript, and there are several possible explanations (genetic background etc). Nonetheless, it would be

helpful for the authors to acknowledge this point in their manuscript, and to provide frequent descriptions of size effects, so that readers can appreciate the extent of variability.

<Response to reviewer>

The GeneSwitch system used in Supplementary Fig. 7 (Supplementary Fig. 12 in the current revised manuscript) is known to have leaky GAL4 activity in the absence of RU486 (Ke and Hsu, 2019, G3; Scialo et al., 2016, PLOS one). The reduced magnitude of GSC loss during aging in 2261>armS10 without RU486 induction may be caused by the leaky expression of armS10; nevertheless, GSC number is significantly increased when adding RU486 for one week. We have added a brief description of the leaky driver to the main text, pg. 11 line 1, and in the figure legend of Supplementary fig. 12d.

5. Supplementary data: For this reviewer, the possible links to Alzheimer's models is not particularly relevant to the report, and seems distracting and premature. I would encourage the authors to consider removing these supplemental data so that they can focus on an interesting GSC phenotype.

<Response to reviewer>

We understand the reviewer's concern. Although these results are not highly related to GSCs, we believe this finding "retrotransposon-GSK3 regulation" may benefit researchers interested in aging-associated tissue degeneration (e.g. AD). In the revised manuscript, we therefore still kept these findings in the supplementary information.

Reviewers' comments:

Reviewer #1 (Remarks to the Author):

The revised manuscript of Lin et al. removed data about cytoplasmic role of Piwi, demonstrated new rescue by constitutively expressing Piwi in the niche, conducted transposon analysis in sorted niche cells, and added new evidence of toll pathway activation. The manuscript continues to present interesting observations of aging-dependent loss of Piwi expression in niche cells and its contribution to GSC loss. While a decrease in Piwi expression correlates with toll activation and impacts on the cadherin/catenin/actin axis, in my opinion, the results shown in the manuscript indicate a correlation between two processes rather than establish a direct causal link.

Major concerns:

1. A direct link between Toll and transposon remains weak.

In rebuttal letter the authors admit that they are not able to provide molecular evidence for a direct link between retrotransposons and Toll signaling and write that they 'soften' the title and changed the abstract accordingly. Honestly, I do not see how the change of the title from "Transposon-activated Immune Signaling in the Aged Niche Eliminates Germline Stem Cells" to "Retrotransposon Derepression in the aged niche Eliminates Germline Stem Cells via Toll-mediated signaling" softens anything. Similarly, the claims of a direct link between Toll and transposons remained in the abstract as authors write that 'retrotransposons in the aged GSC niche act through Toll' and 'retrotransposons also activated GSK3 in fly and human-cell models of Alzheimer's disease.' I simply do not see sufficient experimental evidence for these claims and I believe my concerns are shared by reviewer #3.

While it might be difficult to do in vivo experiments to establish the direct causal relationship between Toll pathway and transposons, it might be possible to do experiments fast in cultured cells. Will expressing gypsy retrotransposon in S2 cells activate Toll?

The authors concluded that transposon and toll work in the same pathway based on rescue by double knock-down experiments. However, no data about the effect of Toll KD alone are presented. Note that in Fig.4b', based on unnormalized GSC counts show that double KD of piwi and toll (3rd column) led to better GSC maintenance compared to control (1st column)! What does it mean? One can envision a scenario where Toll KD alone offers benefits for GSC maintenance, which acts independently of Piwi. This alternative explanation needs to be ruled out.

2. More careful review reveals inconsistencies between effects observed in two piwi RNAi lines.

In Fig.1d, piwi-RNAi-2 on Day 1 (presumably no KD yet) possesses a strangely higher baseline of GSC number (7th column) than ctrl-RNAi or piwi-RNAi-1. The difference in GSC number between control and piwi-RNAi-2 at two weeks appears because of this high baseline value observed piwi-RNAi-2 on Day 1.

Based on unnormalized GSC number counts, piwi-RNAi-2 and control seem to have similar numbers of GSC counts at two weeks indicating that piwi-RNAi-2 might not work at all. Throughout the paper (Fig.2,3,4), all gypsy KD experiments were done with piwi-RNAi-2, whereas all other experiments were done with piwi-RNAi-1. Authors need to justify the usage of different piwi-RNAi lines for different experiments, and whether piwi-RNAi-2 is reliable at all.

In Fig.2e, 5th and 7th columns should be merged, if they are identical experiments. If 7th column is identical to Fig.1d 8th column (as claimed in the figure legend), was it done in parallel with co-KD copia/gypsy? If not, authors need to perform the control side-by-side with experimental groups, given its importance for the claim.

3. The special role of gypsy implied by authors is unconvincing. One of the main conclusions of the paper that single retrotransposon, gypsy, is responsible for the loss of GSC upon Piwi KD.

However, according to authors new results, Piwi KD causes 1.2 fold increase in gypsy mRNA based on RNA-seq data in sorted niche cells. This is very modest increase especially compared to other transposons that were activated as much as 47-fold. I would not trust a 1.2-fold change in RNA-

seq data without several replicas, but even if we believe this result it is rather unexpected that a 1.2-fold change in transposon expression can cause serious effects. This concern is enhanced by the fact that, in Fig.2e, double KD of gypsy and piwi appears to have similar numbers of GSC with piwi-RNAi-2 alone (7th vs. 9th column) indicating that gypsy knockdown has no effect. The difference in two metrics authors present (% of remaining GSC vs. unnormalized number of GSC) hampers claimed the importance of gypsy.

4. It remains unclear why anti-RT drug 3TC reduces viral like particles (VLPs). While authors reasoned that 3TC prevents transposition/expansion of retrotransposons and thus reduces genetical materials for assembling VLPs, they did not detect transposition of gypsy (line 214-5). If the copy number of gypsy did not increase, it is puzzling how inhibiting reverse transcription can reduce VLP assembly? Authors did not show whether VLPs decrease in gypsy KD, either, pointing to a possibility that VLPs come from non-gypsy transposons.

Generally, 3TC is nucleotide analog and while it is known to inhibit reverse transcriptase it likely has effect on other cellular functions. In the context of present study, 3TC might be acting in GSC directly, independently of a niche function. These caveats weaken the usage of 3TC in proving the role of retrotransposon working in a niche to maintain GSC during aging, a claim that requires further support.

5. Alzheimer's data appears out of place and carries weight heavier than authors' evidence. As mentioned by reviewer 3, AD data seems premature. While it ok to keep these results as supplementary figures they should not be used for strong claims in the Abstract.

Other points:

6. gypsy mRNA increased 1.2 fold in line 98 but was wrongly colored to have more than 1.25-fold increase in Fig.2c.

7. Constitutive over-expression of Piwi in niche cells slowed, but did not prevent, GSC loss (supp fig 11). It should be described in text, possibly around line 187 or 200, that there is Piwi-independent maintenance of GSC.

Reviewer #2 (Remarks to the Author):

In the second submitted version of the manuscript entitled "Retrotransposon Derepression in the Aged Niche Eliminates Germline Stem Cells via Toll-mediated Signaling" the authors describe a novel "quality assurance system" for *Drosophila* adult female germ line stem cells. As a symptom of aging, the Piwi transposon regulation system weakens in the stem cell niche. As a consequence, retrotransposons are deregulated and produce virus-like particles capable of infecting the nearby stem cells. De-repression of retrotransposons, however induces a Toll-mediated immune response which activates Glycogen synthase kinase. Glycogen synthase kinase then induces degradation of beta-catenin, which finally results in delocalisation of E-Catherine from CpC-GSC junction. The end result is elimination of GSCs from the stem cell niche infected with the virus-like particles. The focus of the manuscript is the *Drosophila* female stem cell niche. However, the authors present a series of experiments demonstrating that the immune response to retrotransposon de-repression may be a general protective phenomenon. They showed that 1) Piwi and Armadillo expressions are reduced while expression of copia-lacZ is increased in the aged *Drosophila* male GSC niche. 2) 3TC treatment suppresses GSK3 activity in human cell model of Alzheimer's disease. 3) Using the fly Alzheimer model they proved that 3TC treatment partially suppresses GSK3 activity induced in the Ab42-overexpressing fly eyes. In summary, the results presented in the manuscript may be of interest to a wide range of readers.

I made a number of critical remarks about the originally submitted version of the manuscript. In their rebuttal letter, however, the authors gave convincing, acceptable and detailed response to all

my critical comments. The changes that have been made on the manuscript are appropriate and acceptable to me. Accordingly, I propose the revised manuscript to be published.

Reviewer #1 (Remarks to the Author):

Major concerns:

1. *A direct link between Toll and transposon remains weak.*

In rebuttal letter the authors admit that they are not able to provide molecular evidence for a direct link between retrotransposons and Toll signaling and write that they 'soften' the title and changed the abstract accordingly.

Honestly, I do not see how the change of the title from "Transposon-activated Immune Signaling in the Aged Niche Eliminates Germline Stem Cells" to "Retrotransposon Derepression in the aged niche Eliminates Germline Stem Cells via Toll-mediated signaling" softens anything.

Similarly, the claims of a direct link between Toll and transposons remained in the abstract as authors write that 'retrotransposons in the aged GSC niche act through Toll' and 'retrotransposons also activated GSK3 in fly and human-cell models of Alzheimer's disease.' I simply do not see sufficient experimental evidence for these claims and I believe my concerns are shared by reviewer #3.

While it might be difficult to do in vivo experiments to establish the direct causal relationship between Toll pathway and transposons, it might be possible to do experiments fast in cultured cells.

Will expressing gypsy retrotransposon in S2 cells activate Toll?

The authors concluded that transposon and toll work in the same pathway based on rescue by double knock-down experiments. However, no data about the effect of Toll KD alone are presented. Note that in Fig.4b', based on unnormalized GSC counts show that double KD of piwi and toll (3rd column) led to better GCS maintenance compared to control (1st column)! What does it mean? One can envision a scenario where Toll KD alone offers benefits for GSC maintenance, which acts independently of Piwi. This alternative explanation needs to be ruled out.

<Response to the Reviewer>

We thank the reviewer for these comments and suggestions. It has been shown that *gypsy* is silenced by endogenous siRNA in somatic cells like S2 cells (Ghildiyal et al., 2008, Science). Therefore, S2 cells are unlikely to be a workable platform for experiments testing if *gypsy* activates Toll signaling.

Nevertheless, we have performed co-knockdown of *piwi* and *gypsy* in the niche, and we also treated flies carrying the *piwi*-KD niche with 3TC to suppress reverse transcription (which is responsible for generation of virus cDNA). In both of these experiments, we examined Toll-mediated immune signaling, as revealed by the degradation of Cactus (orthologue of mammalian I κ B). None of the experimental

conditions showed rescue of Cactus degradation. Thus, neither expression of *gypsy* alone nor virus cDNA are necessary to cause Toll-mediated Cactus degradation in the *piwi*-KD niche. Furthermore, co-knockdown of *piwi* and *toll* in the niche also did not suppress Cactus degradation. This result suggests that other Toll receptors may be involved in triggering Cactus degradation. We have now added these results in Fig. 4d. and described the results in pg. 14, line 15.

However, niche-specific co-knockdown of *piwi* with either *gypsy*, *toll* or *toll5*, or suppression of reverse transcription in the *piwi*-KD niche by 3TC treatment were all able to reduce GSK3 activity, prevent β -catenin degradation, restore E-cadherin expression in the GSC-niche junction, and rescue GSC number. These results clearly show that retrotransposon cDNA, *gypsy* and Toll-GSK3 signaling are all required to produce the major *piwi*-KD phenotype.

In addition, the data for Toll-KD alone is shown in the Supplementary Table 1. We added a brief description in the main text, pg.13, line 15. Knockdown of *toll* or *toll7* alone in the niche does not affect GSC maintenance as compared to controls, but knockdown of *toll5* alone significantly enhances GSC loss from D1 to 2 weeks after eclosion (33% loss), as compared to controls (12-17% loss, $P < 0.001$). However, co-knockdown of *piwi* with *toll* or *toll5* significantly restores GSC maintenance rate to control levels. These results indicate that Toll-mediated signaling is involved in the GSC loss induced by Piwi depletion in the niche.

In the original fig. 4d, for the grouping ‘% of germaria carrying indicated GSC number’ we only showed the results at 2 weeks. To avoid confusion, we have now also added the results from newly enclosed flies (D1) and show these data in Fig. 4a’ of the revised manuscript. The results show that knockdown of *piwi* in the niche significantly accelerates GSC loss from D1 to 2 weeks after eclosure, while co-knockdown of *piwi* with *toll* or *toll5* in the niche rescues the increased rate of GSC loss induced by *piwi* depletion.

These genetic results clearly support our conclusion that Toll is necessary for the GSC loss phenotype. However, as the reviewer has commented, we did not overexpress retrotransposons in the niche and examine Toll-GSK3 signaling to show a direct connection between these two determinants. We have now changed our Title from the original one implicating retrotransposons to “Piwi Reduction in the Aged Niche Eliminates Germline Stem Cells via Toll-GSK3 Signaling”. We also took out statements in the Abstract that suggest we have direct evidence that retrotransposons are responsible for activating Toll-mediated signaling. Instead, we conclude that in the *piwi*-knockdown niche, retrotransposon derepression is coincident with the activation of Toll-GSK3 signaling for GSC loss. Further, we also

modified our model (Fig. 6b) by adding a question mark (?) to the arrows leading from transposon-derived viral materials to Toll receptors.

2. More careful review reveals inconsistencies between effects observed in two piwi RNAi lines.

In Fig.1d, piwi-RNAi-2 on Day 1 (presumably no KD yet) possesses a strangely higher baseline of GSC number (7th column) than ctrl-RNAi or piwi-RNAi-1. The difference in GSC number between control and piwi-RNAi-2 at two weeks appears because of this high baseline value observed piwi-RNAi-2 on Day 1.

Based on unnormalized GSC number counts, piwi-RNAi-2 and control seem to have similar numbers of GSC counts at two weeks indicating that piwi-RNAi-2 might not work at all. Throughout the paper (Fig.2,3,4), all gypsy KD experiments were done with piwi-RNAi-2, whereas all other experiments were done with piwi-RNAi-1.

Authors need to justify the usage of different piwi-RNAi lines for different experiments, and whether piwi-RNAi-2 is reliable at all.

In Fig.2e, 5th and 7th columns should be merged, if they are identical experiments. If 7th column is identical to Fig.1d 8th column (as claimed in the figure legend), was it done in parallel with co-KD copia/gypsy? If not, authors need to perform the control side-by-side with experimental groups, given its importance for the claim.

<Response to Reviewer>

We thank the reviewer for these comments and questions.

Regarding Fig. 1d: As the reviewer noted, *bab1>piwiRNAi-2* germaria carried more GSCs at Day 1 than controls and *bab1>piwiRNAi-1* flies. Variation such as this is a common observation in experimental lines, probably based on genetic backgrounds. We have now added a short statement in the main text to alert readers to this fact, pg.4, line 15-18. However, GSC loss rates in the two independent *piwi* RNAi lines are similar from D1 to 2 weeks (*piwiRNAi-1*: 34% versus *piwiRNAi-2* 32% GSC loss) and from 2 to 5 weeks (*piwiRNAi-1*: 57% versus *piwiRNAi-2* 53% GSC loss).

Like *piwiRNAi-1*, *bab1>piwiRNAi-2* niches also show nearly complete absence of Piwi. We have now added this result to Fig. 1c, and shortly describe it in the main text, pg 4, line 4-6. In addition, this *piwiRNAi-2* line (Bloomington Drosophila Stock Center #33724) has been used in several studies (e.g., Jin et al., 2003, Current Biology); its efficiency in disrupting *piwi* expression has been validated.

The main reason we used *piwiRNAi-2* (on the third chromosome) for the *gypsy* knockdown experiments is because both *piwiRNAi-1* and *gypsyRNAi* are located on the second chromosome. We did not generate a recombination line of *piwiRNAi-1*

and *gypsyRNAi* lines, as it would be too difficult to validate the presence of both *RNAi* inserts in candidate flies. We have now added short explanation for why we used *piwiRNAi-2* in the Materials and Methods, pg. 20, line 18.

Regarding Fig. 2e: The original 5th and 7th columns cannot be merged, as the food conditions were different. As described in the Materials and Methods (pg.22, line 5), flies for 3T experiments were fed with wet yeast with or without 3TC. For knockdown experiments (pg. 20, line 17), flies were fed with normal diet with dry yeast powder.

For the original 8th column, the experiment was not done in parallel with co-KD *copia/gypsy*. We have now added the results of *piwiRNAi-2* performed in parallel with co-KD *gypsy* experiments. Both of the results (*bab1>piwiRNAi-2* alone in Fig. 1d and Fig. 2e) are listed in Supplementary Table 1 and the two experiments displayed similar rates of GSC loss.

3. The special role of gypsy implied by authors is unconvincing.

One of the main conclusions of the paper that single retrotransposon, gypsy, is responsible for the loss of GSC upon Piwi KD.

However, according to authors new results, Piwi KD causes 1.2 fold increase in gypsy mRNA based on RNA-seq data in sorted niche cells.

This is very modest increase especially compared to other transposons that were activated as much as 47-fold. I would not trust a 1.2-fold change in RNA-seq data without several replicas, but even if we believe this result it is rather unexpected that a 1.2-fold change in transposon expression can cause serious effects.

This concern is enhanced by the fact that, in Fig.2e, double KD of gypsy and piwi appears to have similar numbers of GSC with piwi-RNAi-2 alone (7th vs. 9th column) indicating that gypsy knockdown has no effect. The difference in two metrics authors present (% of remaining GSC vs. unnormalized number of GSC) hampers claimed the importance of gypsy.

<Response to Reviewer>

We understand the reviewer's concern. As we describe in the main text (pg. 6, line 5), the niche cells we isolated for RNA sequencing contained cap cells and terminal filament (TF) cells, which express very low levels of Piwi but high levels of *gypsy* transcripts (examined by *in situ* hybridization). The presence of TF cells with high *gypsy* transcripts may explain this modest increase of *gypsy* in the RNA sequencing result. We have now added our *in situ* hybridization result to Supplementary Fig. 2, and added a description in the main text, pg. 5, line 13, and pg.

6, line 14.

In the original fig. 2e grouping ‘% of germaria carrying indicated GSC number’ we only showed the results at 2 weeks. To avoid confusion, we have now also added results from newly enclosed flies (D1) in the revised manuscript. The results show that *piwi*-KD and *piwi* & *copia* co-KD niches show similar rates of GSC loss from D1 to 2 weeks (*piwi*-KD, 31%; *piwi* & *copia* co-KD, 29%), while *piwi* & *gypsy* co-KD niches only display 18% GSC loss, comparable to that in *gfp*-KD control niche with or without 3TC treatment. Our results clearly show that suppressing *gypsy* expression in the *piwi*-KD niche rescues the GSC loss phenotype induced by *piwi* depletion in the niche.

4. It remains unclear why anti-RT drug 3TC reduces viral like particles (VLPs). While authors reasoned that 3TC prevents transposition/expansion of retrotransposons and thus reduces genetical materials for assembling VLPs, they did not detect transposition of gypsy (line 214-5). If the copy number of gypsy did not increase, it is puzzling how inhibiting reverse transcription can reduce VLP assembly? Authors did not show whether VLPs decrease in gypsy KD, either, pointing to a possibility that VLPs come from non-gypsy transposons.

Generally, 3TC is nucleotide analog and while it is known to inhibit reverse transcriptase it likely has effect on other cellular functions. In the context of present study, 3TC might be acting in GSC directly, independently of a niche function. These caveats weaken the usage of 3TC in proving the role of retrotransposon working in a niche to maintain GSC during aging, a claim that requires further support.

<Response to Reviewer>

We thank the reviewer for raising this point and agree with the comment that although VLPs accumulate in *piwi*-KD niches, those VLPs may not be generated by *gypsy*. *piwi*-KD niches treated with 3TC exhibit decreased VLPs when compared to *piwi*-KD niche without 3TC treatment, suggesting that VLPs may be related to retrotransposon cDNA made via reverse transcription. Notably, VLPs can also be directly generated from retrotransposon transcripts present in the cytoplasm. Because *gypsy* transposition was not found in our genomic sequencing results, the VLPs we observed may not be formed by *gypsy*. Indeed, our new experiments show that co-knockdown of *gypsy* and *piwi* cannot reverse VLP accumulation. We have now added this data to Fig. 5a, and shortly described it in the main text, pg. 15, line 6.

We also cannot rule out the possibility that 3TC may have effects on other cellular functions, or somehow act on GSC directly. However, GSC numbers are

similar in the *gfp*-KD flies with or without 3TC treatment, indicating that GSC maintenance is not affected by 3TC treatment for at least the two-week experimental period. However, same treatment in flies with *piwi*-KD niche reverses the GSC loss phenotype, including effects on GSK3 activity, beta-catenin degradation and E-cadherin expression in the GSC-niche junction that are observed in the *piwi*-KD niche. Overall our experiments, including fly and human cell models for Alzheimer's disease, suggest retrotransposon derepression is highly relevant to age-related diseases, and 3TC treatment may be beneficial for these diseases.

5. Alzheimer's data appears out of place and carries weight heavier than authors' evidence. As mentioned by reviewer 3, AD data seems premature. While it ok to keep these results as supplementary figures they should not be used for strong claims in the Abstract.

<Response to Reviewer>

We thank the reviewer and agree with the comment. We have now completely removed AD information from the Abstract.

6. gypsy mRNA increased 1.2 fold in line 98 but was wrongly colored to have more than 1.25-fold increase in Fig.2c.

<Response to Reviewer>

We thank the reviewer for pointing out this mistake; we have now corrected it from 1.2-fold change to 1.26-fold change in pg. 6, line 13.

7. Constitutive over-expression of Piwi in niche cells slowed, but did not prevent, GSC loss (supp fig 11). It should be described in text, possibly around line 187 or 200, that there is Piwi-independent maintenance of GSC.

<Response to Reviewer>

We thank the reviewer for this comment. We have modified our sentence and added short note that GSC loss during aging also occurs via Piwi-independent mechanisms, pg. 11, line 9 and pg. 12, line 4.

REVIEWERS' COMMENTS:

Reviewer #1 (Remarks to the Author):

The authors response to reviewers concerns is reasonable to me.

The authors performed multiple new experiments and reported what they found, regardless of whether those new results supported their hypothesis or not. They have also softened their statements in the manuscript, based on what the new experiments told them.